# KSRP specifies monocytic and granulocytic differentiation through regulating miR-129 biogenesis and RUNX1 expression

Hongmei Zhao[1,2], Xiaoshuang Wang[1], Ping Yi[3], Yanmin Si[1], Puwen Tan[4], Jinrong He[1], Shan Yu[1], Yue Ren[1], Yanni Ma[1], Junwu Zhang[1], Dong Wang [4,5], Fang Wang[1] & Jia Yu[1]

RNA-binding proteins (RBPs) integrate the processing of RNAs into post-transcriptional gene regulation, but the direct contribution of them to myeloid cell specification is poorly understood. Here, we report the first global RBP transcriptomic analysis of myeloid differentiation by combining RNA-seq analysis with myeloid induction in CD34[+] hematopoietic progenitor cells. The downregulated expression of the KH-Type Splicing Regulatory Protein (KSRP) during monocytopoiesis and up-regulated expression during granulopoiesis suggests that KSRP has divergent roles during monocytic and granulocytic differentiation. A further comparative analysis of miRNA transcripts reveals that KSRP promotes the biogenesis of miR-129, and the expression patterns and roles of miR-129 in myeloid differentiation are equivalent to those of KSRP. Finally, miR-129 directly blocks the expression of Runt Related Transcription Factor 1 (RUNX1), which evokes transcriptional modulation by RUNX1. Based on our findings, KSRP, miR-129, and RUNX1 participate in a regulatory axis to control the outcome of myeloid differentiation.

[1] State Key Laboratory of Medical Molecular Biology, Department of Biochemistry & Molecular Biology, Institute of Basic Medical Sciences, Chinese Academy of Medical Sciences (CAMS) & Peking Union Medical College (PUMC), Beijing 100005, China. [2] State Key Laboratory of Medical Molecular Biology, Department of Physiology and Pathophysiology, Institute of Basic Medical Sciences, Chinese Academy of Medical Sciences (CAMS) & Peking Union Medical College (PUMC), Beijing 100005, China. [3] Department of Obstetrics and Gynecology, The Third Affiliated Hospital of Chongqing Medical University, Chongqing 401120, China. [4] College of Bioinformatics Science and Technology, Harbin Medical University, Harbin 150081, China. [5] Center for Informational Biology, University of Electronic Science and Technology of China, Chengdu 610054, China. Hongmei Zhao, Xiaoshuang Wang and Ping Yi contributed equally to this work. Correspondence and requests for materials should be addressed to D.W. (email: wangdong@ems.hrbmn.edu.cn) or to F.W. (email: wo_wfang@hotmail.com) or to J.Y. (email: j-yu@ibms.pumc.edu.cn)

Myeloid cell differentiation requires the timely regulation of gene expression for the acquisition of the mature blood cell phenotype, and properly differentiated myeloid cells are essential for normal immune responses[1]. In particular, granulocytes, monocytes and macrophages are crucial to defend against intruding pathogens and to recognize cellular molecules released by damaged tissues[2]. The myeloid lineage-specific pattern of gene expression depends on the interplay of a variety of elements, including transcription factors, epigenetic mechanisms and multiple post-transcriptional regulators. Lineage-specifying transcription factors include PU.1, which affects the hematopoiesis of multiple lineages in an early stage[3], and Gfi-1, C/EBPα and C/EBPε, which govern the differentiation of hematopoietic stem cells (HSCs) along the myeloid lineage toward granulocytes rather than monocytes[4,5]. Moreover, epigenetic controls, such as histone modification and DNA methylation, are also involved. Although many factors are involved in the differentiation of granulocyte-monocyte progenitors (GMPs) into granulocytes and monocytes, only PU.1 and C/EBPε were confirmed to favor the commitment to one lineage over another[6].

Over the last decade, knowledge of the network of regulatory circuits incorporating RNA-binding proteins (RBPs) and non-coding RNAs (ncRNAs) has accumulated, adding another complexity to post-transcriptional regulation. RBPs could regulate many aspects of RNA processing, including alternative splicing, RNA transport and stability, RNA localization and mRNA translation[7,8]. By binding to RNAs, RBPs play positive (activators) or negative (repressors) roles, depending on the protein, the RNA and the biological context[9]. Alterations in expression or mutations in RBPs or their binding sites in target transcripts have been reported to cause several human diseases, including genetic diseases, neurological disorders and cancer[10–14]. Furthermore, an expanding set of RBPs has been shown to be related to multiple hematopoietic malignancies[15,16]. However, the direct contribution of RBPs to myeloid cell specification is poorly understood.

Here we used a set of genetic and bioinformatic tools to dissect how RBPs control myelopoiesis. We focused on an RBP, KSRP, to characterize its roles during monocytic and granulocytic differentiation and identified its target RNA transcripts. For the first time, KSRP is shown to be differentially regulated during human monocytic and granulocytic differentiation. Attenuation of KSRP is required for monocyte differentiation while enhanced KSRP expression is indispensable for granulocyte differentiation. Subsequently, KSRP interacts with pri-miR-129 transcript, promotes its post-transcriptional processing and enhances the miR-129 biogenesis. Thus, miR-129 acts downstream of KSRP to regulate myeloid differentiation and targets RUNX1. Based on these data, the KSRP−miR-129-RUNX1 regulatory axis promotes granulocyte differentiation at the expense of monocyte-macrophage differentiation, suggesting its role in orchestrating the ratio of these two phagocytes.

## Results

**Identification of RBPs involved in myeloid differentiation.** Monocytic (M-) or granulocytic (G-) differentiation of CD34+ HPCs was induced by treating the cells with macrophage colony-stimulating factor (M-CSF) or granulocyte colony-stimulating factor (G-CSF), respectively, for a 15-day period (Fig. 1a, left)[17]. Monocytes appeared at day 5, and the number of macrophages increased at days 10 and 15 of monocyte differentiation. Meanwhile, the proportion of metamyelocytes exceeded the proportion of promyelocytes at day 10, and neutrophilic band cells were observed at day 15 of granulocyte differentiation (Fig. 1a, right). We performed a global RNA-seq analysis of RNA isolated from cells obtained at 5, 10, and 15 days of differentiation to characterize the changes in RBP genes during M or G differentiation (Fig. 1b, Supplementary Fig. 1a). Two independent biological replicates were assessed, and the comparisons of the gene expression levels in duplicate samples indicated good reproducibility (Supplementary Fig. 1b).

To investigate the RBP expression patterns, we first performed a differential expression (DE) analysis of Ensembl-annotated human genes by EBSeq-HMM[18] (Supplementary Data 1). Combined with RBPDB- and ATrRACT-annotated[19,20] human RBP genes, the DE analysis revealed 1,100 (from batch 1) and 1,188 (from batch 2) monotonically up-regulated or down-regulated genes, including 21 (from batch 1) and 16 (from batch 2) RBP genes (Fig. 1c and Supplementary Data 2). K-means clustering of the expression data from the intersections of batch 1 and batch 2 revealed seven distinct clusters of expression of 646 genes, with 8 RBPs located in clusters 2, 3, 5, and 7 (Fig. 1c, d and Supplementary Data 3).

The 8 differentially expressed RBP genes at the intersection of the two batches that are considered as candidates regulating M or G differentiation were analyzed to identify the key RBP genes. Among these genes, 4 were downregulated during monocyte differentiation, 2 were downregulated during granulocyte differentiation and 2 were up-regulated during granulocyte differentiation (Fig. 1e and Supplementary Data 4). qPCR was performed to validate these RNA-seq results in a 20-day differentiation culture. As a quality control, we examined the expression of known lineage-specific marker genes and found that the levels of both monocyte differentiation markers (*M-CSFR*, *CD14*, *ITGAM*, and *SP1*) and granulocyte differentiation markers (*ITGAM*, *G-CSFR*, *CEBPA*, and *MPO*) were similar to previous reports (Fig. 1f).

As shown in Fig. 1g, the fold changes in gene expression were quite consistent with their expression level determined by RNA-seq. Namely, *YBX1*, *KSRP*, *U2AF2*, *ZC3H12A*, *TTC14*, and *HNRNPAIP7* exhibited a consistent change in expression compared to the RNA-seq data. Among these RBPs, *KSRP* drew our interest because it exhibited a dramatic increase in expression at a later stage (20 days) of granulocytic differentiation, which is opposite to its temporal expression pattern observed during monocytic differentiation (Fig. 1g).

**KSRP is reciprocally expressed in M and G differentiation.** To confirm the expression pattern of the *KSRP* mRNA, Western blotting was performed in cord blood CD34+ HPCs (Fig. 2a). The levels of the KSRP protein were gradually downregulated during M-CSF-induced monocytic differentiation, but dramatically upregulated during G-CSF-induced granulocytic differentiation (Fig. 2a; and Supplementary Fig. 2a). Similar expression tendency of KSRP was also observed in bone marrow derived CD34+ HPCs, the normal source of monocytes and granulocytes in adults (Fig. 2b; and Supplementary Fig. 2b). Meanwhile, three myeloid leukemia cell lines, THP− 1, NB4 and HL-60, were also investigated. The levels of the KSRP mRNA and protein were decreased in THP− 1 and HL-60 cells upon phorbol myristate acetate (PMA)-induced monocytic differentiation but increased in NB4 and HL-60 cells upon all-trans retinoic acid (ATRA)-induced granulocytic differentiation (Fig. 2c, d). Thus, KSRP might have opposite roles during monocytic and granulocytic differentiation.

**KSRP has divergent roles in M and G differentiation.** To investigate the functional significance of KSRP in myeloid differentiation, lenti-scramble- or lenti-sh-KSRP− transduced cord blood HPCs undergoing M and G differentiation, respectively, were analyzed. As expected, KSRP silencing in HPCs increased the expression of the *CD14* mRNA beginning on day 5 of

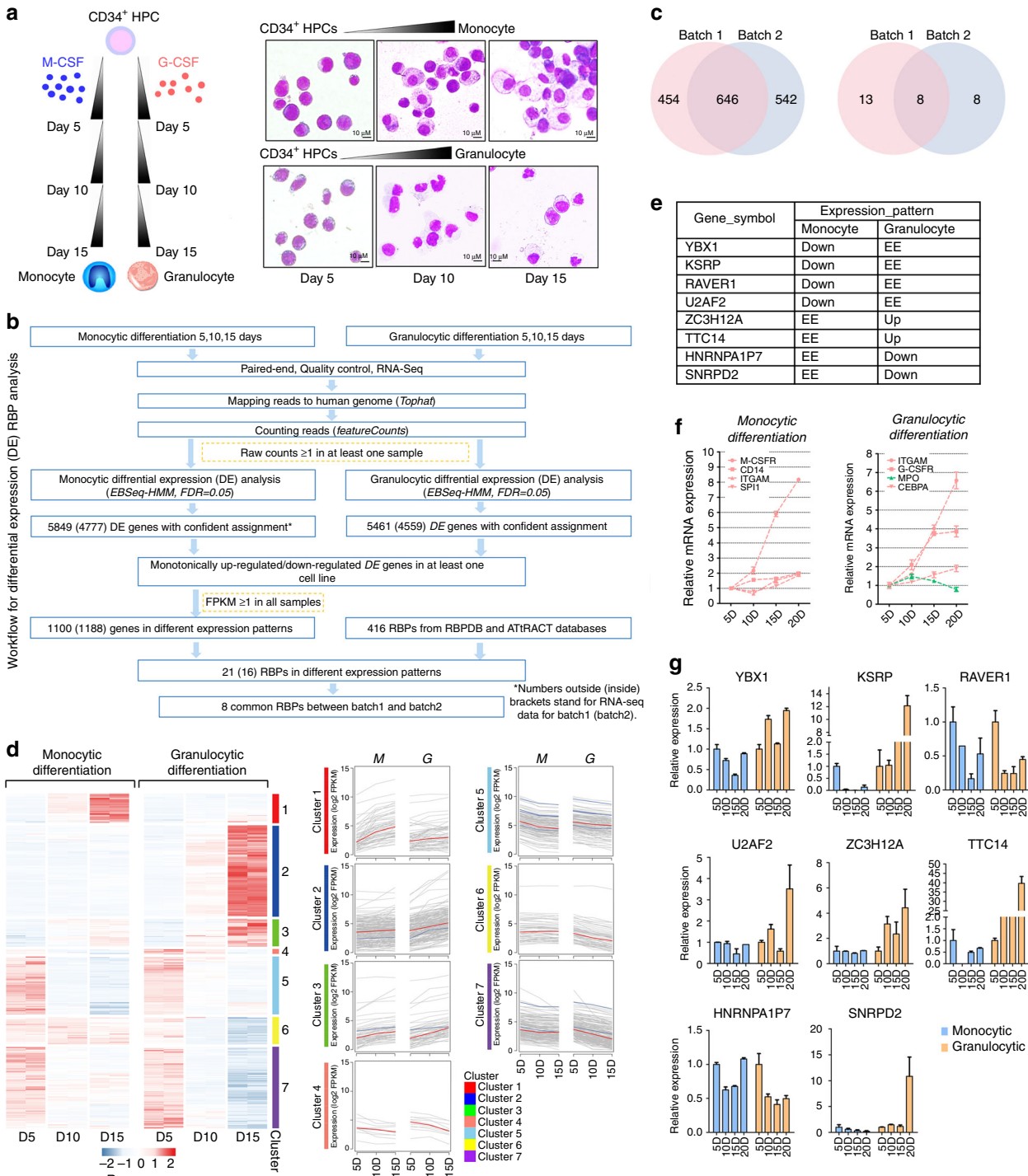

**Fig. 1** RNA-seq identifies differentially expressed RBPs. **a** A schematic representation of the procedure used to prepare the samples for the RNA-seq analysis (left). CD34+ HPCs obtained from cord blood were differentiated into either the monocyte or granulocyte lineage, and cells were collected at 5, 10 and 15 days for the RNA-seq analysis. Temporal morphological changes (May-Grunwald Giemsa staining) in differentiated HPCs (Right). Magnification, 40×. **b** The workflow for differential expression (DE) RBP analysis. **c** Overlap of the global differentially expressed genes in **b** (left) or differentially expressed RBP genes (right) between batch 1 and batch 2. **d** Left panel: heat map showing temporal expression of genes during monocytic or granulocytic differentiation of CD34+ HPCs in duplicate samples from batch 1 and batch 2. Right panel: K-means clustering of the temporal profiles of the differentially expressed genes (DEGs) divided into seven individual clusters. RBP genes are shown as blue lines. **e** RNA-seq data for potential RBP genes that regulate myeloid differentiation. 'Up' or 'Down' represent a gene that is monotonically up-regulated or downregulated in the corresponding culture, respectively, whereas 'EE' indicates a gene that is not a DEG in this cell culture. **f** qPCR analysis of *M-CSFR, CD14, ITGAM* and *SP1* mRNA expression during monocyte differentiation and *ITGAM*, G-CSFR, *CEBPA* and *MPO* mRNA expression during granulocyte differentiation. Up-regulated genes were shown in red and downregulated gene was shown in green. Three technical replicates from a single experiment. **g** qPCR analysis of the expression of the eight candidate RBP mRNAs during monocyte and granulocyte differentiation. Three technical replicates from a single experiment. Data are shown as means±s.d.

monocyte differentiation (Fig. 2e), and specifically affected the production of CD14[+]/CD11b[+] cells in lentivirus-transduced GFP[+] HPCs (Fig. 2f) and colony-forming units-monocytes (CFU-M) (Fig. 2h) on day 15 of monocyte differentiation. In contrast, granulocytic differentiation was attenuated after KSRP repression, as demonstrated by the lower expression of the *CD11b* mRNA (Fig. 2e), decreased number of CD11b[+] cells in GFP[+] HPCs (Fig. 2g) and fewer colony-forming units-granulocytes (CFU-G) (Fig. 2h). Consistent with these observations, Giemsa staining showed that KSRP knockdown increased the numbers of macrophages on day 16 of monocyte differentiation (Fig. 2f), but obviously decreased the number of band form and segmented neutrophils on day 16 of granulocyte differentiation (Fig. 2g) compared with the lenti-scramble transduced group.

Meanwhile, THP-1 and NB4 cells were transfected with sh-KSRP (Supplementary Fig. 2c), which resulted in ~70%

knockdown of the KSRP levels (Fig. 2i; and Supplementary Fig. 2d) compared with the sh-control (pSIH), and then induced with PMA or ATRA for 48 h. Compared with the control, KSRP knockdown increased the percentage of CD14[+] THP-1 cells and *CD14* mRNA expression; conversely, KSRP knockdown decreased the percentage of CD11b[+] NB4 cells and decreased the *CD11b* mRNA levels (Fig. 2j; Supplementary Fig. 2e–g). Therefore, the loss-of-function analysis in HPCs and myeloid cell lines showed that KSRP has divergent roles during myeloid M- and G-differentiation.

**KSRP regulates miRNA biogenesis in myeloid cells.** KSRP has been shown to regulate mRNA stability, mRNA localization, and mRNA translation in different systems[21–23]. Nevertheless, here we focused on its recently reported roles in miRNA processing

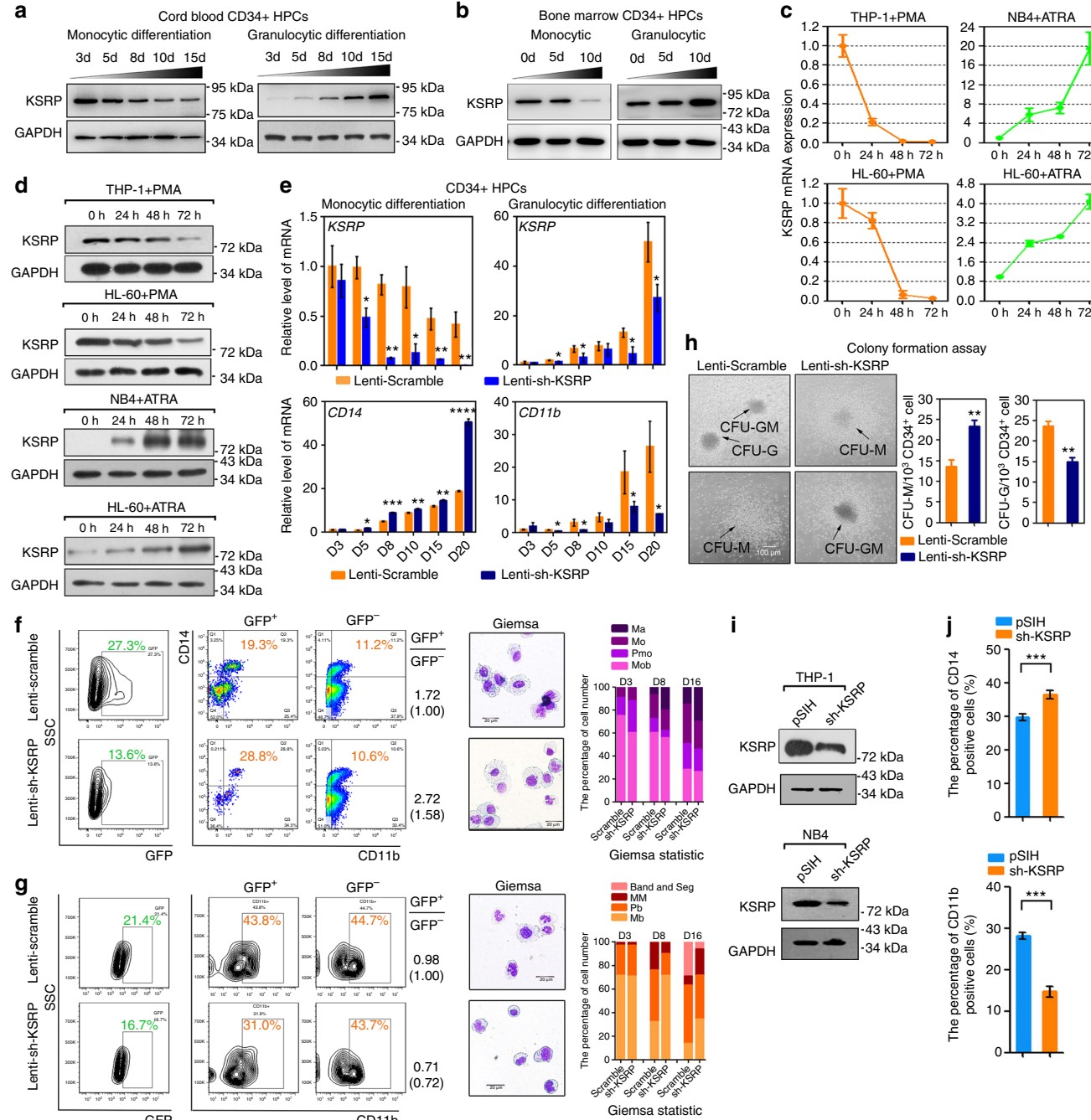

because several miRNAs regulate myeloid differentiation[17,24]. A comparative analysis of miRNA transcripts expressed in KSRP-overexpressing THP-1 cells was performed by poly(A)-enriched RNA sequencing (primary miRNA transcripts) and small RNA sequencing (mature miRNA transcripts) to determine whether KSRP negatively regulates monocyte/granulocyte differentiation by modulating miRNA biogenesis (Fig. 3a). A bioinformatics analysis of the intersection of the changes in the expression of both pri-miRNAs and mature miRNAs identified 16 candidates whose biogenesis was promoted by KSRP, but only 1 candidate that was inhibited by KSRP (Fig. 3b, c and Supplementary Data 5 and 6), confirming the previously reported positive role of KSRP in pri-miRNA processing[25]. The expression of the 16 up-regulated miRNAs was evaluated by qPCR in KSRP-overexpressing or KSRP knockdown THP-1 cells to confirm that KSRP directly mediated the processing of these miRNAs (Fig. 3d). Thirteen (with the exception of miR-140, -33b and -941) of the 16 miRNAs showed decreased expression of the pri-miRNA but increased expression of the miRNA upon KSRP overexpression, and a reverse expression pattern was observed upon KSRP knockdown (Fig. 3e). In addition, we also observed continuous up-regulation of pri-129-1, -615, -98, -941, and -140 and down-regulation of their mature transcripts in THP-1 cells undergoing monocytic differentiation (Supplementary Fig. 3a). However, only pri-129-1 and -140 showed an inverse expression pattern in NB4 cells undergoing granulocyte differentiation (Supplementary Fig. 3a, lower panel), suggesting that these miRNAs are functionally regulated during myeloid differentiation. Based on these results, miR-129-5p was selected for further study.

**KSRP regulates the processing of pri-129-1 in myeloid cells.** Since human *miR-129* is presented on two different genomic loci (*miR-129-1* (7q32.1) and *miR-129-2* (11p11.2)) (Fig. 4a), the transcription efficiencies of these loci were evaluated. The pri-129 transcripts were primarily expressed from the *miR-129-1* locus (Fig. 4a). Likewise, read counts from RNA-seq were mainly mapped to the *miR-129-1* locus but not to the *miR-129-2* (Supplementary Fig. 3b), confirming the transcription bias from *miR-129-1* locus. Therefore, the subsequent analysis focused only on *miR-129-1*. The full-length pri-129-1 transcript was obtained by 5′ and 3′ RACE, and a specific 694-nt RNA transcript (Supplementary Fig. 3c) was identified that perfectly mapped to the *miR-129-1* locus in the human genome. Both qPCR and Northern blotting were performed to further confirm the expression of pri-129-1 and miR-129-5p (hereafter designated miR-129). The

expression of pri-129-1 was increased in PMA-induced THP-1 cells but decreased in ATRA-induced NB4 cells (Fig. 4b; Supplementary Fig. 3d, e). In contrast, the precursor (pre-129-1) and miR-129 displayed expression patterns opposite to the expression of pri-129-1 (Fig. 4b; Supplementary Fig. 3d). The same results were obtained for pri-129-1, pre-129-1 and miR-129 expression during the M and G differentiation of HPCs from cord blood or bone marrow (Fig. 4c, d). We next measure the levels of pri-129-1, pre-129-1 and miR-129 in THP-1 cells in which KSRP was knocked down (pSIH-sh-KSRP) or overexpressed (pEGFP-KSRP) to further validate the regulatory role of KSRP in pri-129-1 processing. As expected, KSRP overexpression increased the levels of pre-129-1 and miR-129, but reduced the levels of pri-129-1, whereas KSRP knockdown led to pri-129-1 reduction, but pre-129-1 and miR-129 increase (Fig. 4e). Therefore, KSRP regulated pri-129-1 processing in myeloid cells.

**KSRP binds the pri-129-1 transcript in myeloid cells.** RNAfold software was used to identify the KSRP-binding sites on the pri-129-1 transcript and investigate the direct interaction of KSRP with pri-129-1. Four KSRP-binding sites were identified (Fig. 5a), all of which were located in the loop region. KSRP has been shown to modulate pri-let-7a processing by binding to its stem-loop region[25], but our observations identified novel interactions between KSRP and sequences far from the pri-129-1 stem-loops. Subsequently, RNA immunoprecipitation (RIP) was performed to validate the predicted physical interactions, and three sets of primers were used to detect pri-129-1 in the RNA precipitates (Fig. 5b). The IP efficiency of the KSRP antibody has been verified (Supplementary Fig. 4a). KSRP bound endogenous pri-129-1 and pri-let-7a but not pri-23b (pri-let-7a and pri-23b were used as the positive and negative controls, respectively, Fig. 5c, d), and the binding of KSRP to pri-129-1 was decreased after PMA treatment (Supplementary Fig. 4b).

This interaction was further tested with an RNA pull-down analysis (Fig. 5e). 293 T cells expressing MS2-tagged wild-type pri-129-1 (129_WT-MS2) in which the MS2 hairpins were captured by the MS2 coat protein expressed ~7000-fold more pri-129-1 than the cells expressing the empty constructs (Supplementary Fig. 4c, left panel). The addition of the MS2-MBP (maltose-binding protein) fusion protein and amylose resin to the lysates recovered up to 10% of pri-129-1 RNAs (Supplementary Fig. 4c). More importantly, KSRP was co-purified with 129_WT-MS2 but not the empty constructs; GAPDH was used as the negative control (Fig. 5f). In addition, in a competitive RNA pull-down experiment, KSRP binding to pri-129-1 was efficiently and

---

**Fig. 2** KSRP exhibits different functions during monocytic and granulocytic differentiation. **a**, **b** Immunoblot analysis of KSRP protein in CD34[+] HPC models of monocytic and granulocytic differentiation from cord blood **a** or bone marrow **b**. **c** qPCR analysis of *KSRP* mRNA in THP-1 and HL-60 cells undergoing monocyte differentiation (left), and in NB4 and HL-60 cells undergoing granulocyte differentiation (right). Three technical replicates from a single experiment representative of two independent experiments. **d** Immunoblot analysis of KSRP expression as indicated in **c**. **e** qPCR detection of the *KSRP*, *CD14* and *CD11b* levels in CD34[+] HPCs transduced with lenti-sh_KSRP or lenti-scramble control during monocytic and granulocytic differentiation. Three technical replicates from a single experiment. **f** CD34[+] HPCs were infected with lenti-sh-KSRP or lenti-scramble, followed by M-CSF stimulation. Left: FACS analysis of the CD14[+]/CD11b[+] cells in GFP[+] and GFP[−] cells; The ratio of CD14[+]/CD11b[+] cells in GFP[+] to that in GFP[−] was shown in the right (parentheses means the normalized ratio to the control); right: May-Grunwald Giemsa staining and the statistical analysis of the percentages of monoblasts (Mob), promonocytes (PMo), monocytes (Mo) and macrophages (Ma). **g** CD34[+] HPCs were infected with lenti-sh-KSRP or lenti-scramble, followed by G-CSF stimulation. Left: FACS analysis of the CD11b[+] cells in GFP[+] and GFP[−] cells; The ratio of CD11b[+] cells in GFP[+] to that in GFP[−] was shown in the right; right: May-Grunwald Giemsa staining and the statistical analysis of the percentages of myeloblasts (Mb), promyelocytes (Pb), metamyelocytes (MM) and band & segmented neutrophils. Magnification, 40×. The bar represents 10 μm. **h** Colony-forming assays of CD34[+] HPCs and the statistical analysis of CFU-M and CFU-G. Three technical replicates from a single experiment. The bar represents 50 μm. **i** Immunoblot analysis of the KSRP protein in THP-1 and NB4 cells transduced with pSIH-sh-KSRP or the pSIH control. **j** Quantitation of the FACS results showing CD14 expression during monocytic differentiation of THP-1 cells transfected with sh-KSRP or pSIH control (upper), and CD11b expression during granulocytic differentiation of NB4 cells transfected with sh-KSRP or pSIH control (lower). Three technical replicates from a single experiment representative of two independent experiments. Data are shown as means±s.d. *$P < 0.05$, **$P < 0.01$, ***$P < 0.001$, Student's *t*-test

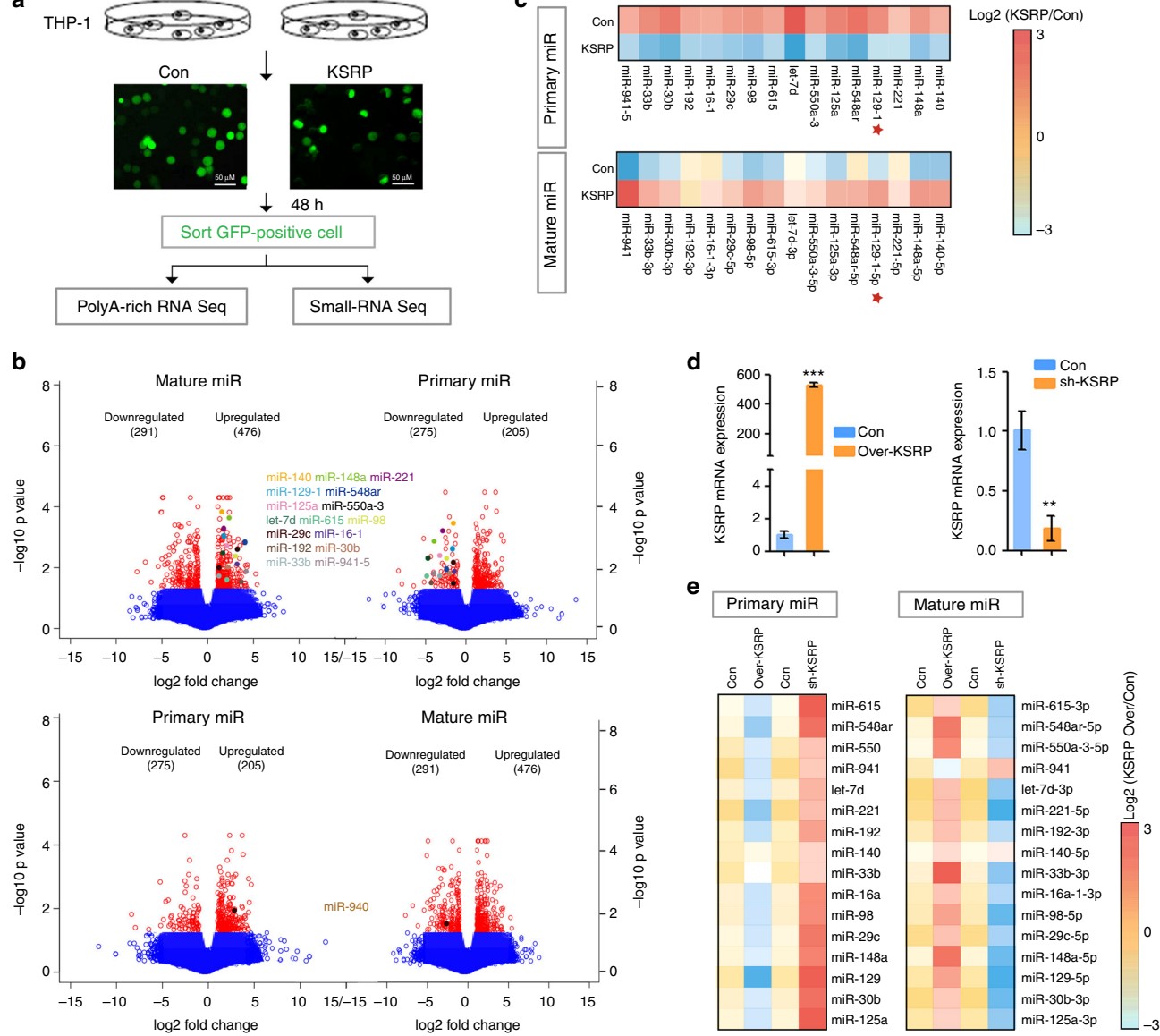

**Fig. 3** RNA-seq identifies KSRP regulated primary miRNA. **a** Microscopy images of THP-1 cells transduced with lentiviruses expressing GFP (Con) or KSRP (KSRP). The GFP+ cells were subsequently sorted by FACS. The bar represents 50 μm. **b** Volcano plots displaying the average log2 ratios of the expression of all pri-miRNAs in poly(A)-enriched RNA-seq data and the expression of mature miRNAs in small RNA-seq data from KSRP versus Con groups obtained in **a**. miRNA candidates whose processing was either promoted (upper) or inhibited (lower) by KSRP were shown in multi-color. miRNAs with statistically significant ($p$-value < 0.05, pairwise multiple comparisons) are shown in red. **c** Heat map representation of relative pri-miRNA and mature miRNA expression levels obtained from the RNA-seq data for the 16 selected miRNAs. **d**, **e** THP-1 cells in which endogenous KSRP was knocked down or overexpressed were analyzed by qPCR for KSRP expression **d** or the primary and mature transcripts expression corresponding to the 16 miRNAs **e**. Three technical replicates from a single experiment. Data are shown as means±s.d. **$P$ < 0.01, ***$P$ < 0.001, Student's $t$-test

competitively inhibited by increasing amounts of non-MS2-tagged pri-129-1 (129_WT) (Fig. 5g; Supplementary Fig. 4d, e), but not non-MS2-tagged pri-23b (Supplementary Fig. 4e). Moreover, the amount of co-precipitated KSRP was accordingly increased with the addition of 129_WT-MS2 (Supplementary Fig. 4f), suggesting the specificity of the interaction.

Five MS2-tagged mutant pri-129-1 constructs containing one or all of the KSRP-binding site mutations were used in an RNA pull-down assay to determine the bona fide binding site for KSRP among the four predicted sites (Fig. 5h). Co-purification with KSRP was attenuated by 129_mut1-MS2, 129_mut2-MS2 and 129_mut4-MS2 and thoroughly abrogated with 129_mutF-MS2 compared with 129_WT-MS2 (Fig. 5h), indicating that sites 1, 2 and 4 were required for KSRP targeting. An RNase H protection

assay was then performed by incubating RNAs harboring predicted KSRP-binding sites 1 and 2 (clone 1) or 3 and 4 (clone 2) with their complementary DNA probes to confirm this hypothesis; a probe complementary to an irrelevant sequence was used as a negative control (Fig. 5i). The preincubation with KSRP reduced the RNase H-mediated digestion of the samples directed by probe 1, 2 and 4, but not probe 3 (Fig. 5j), indicating the protective effects of KSRP on sites 1, 2, and 4 (Fig. 5k). On the basis of these results, KSRP specifically binds to three of the four predicted sites in pri-129-1.

**KSRP promotes processing of pri-129-1 in vitro.** Since KSRP forms complexes with Drosha to regulate the biogenesis of a

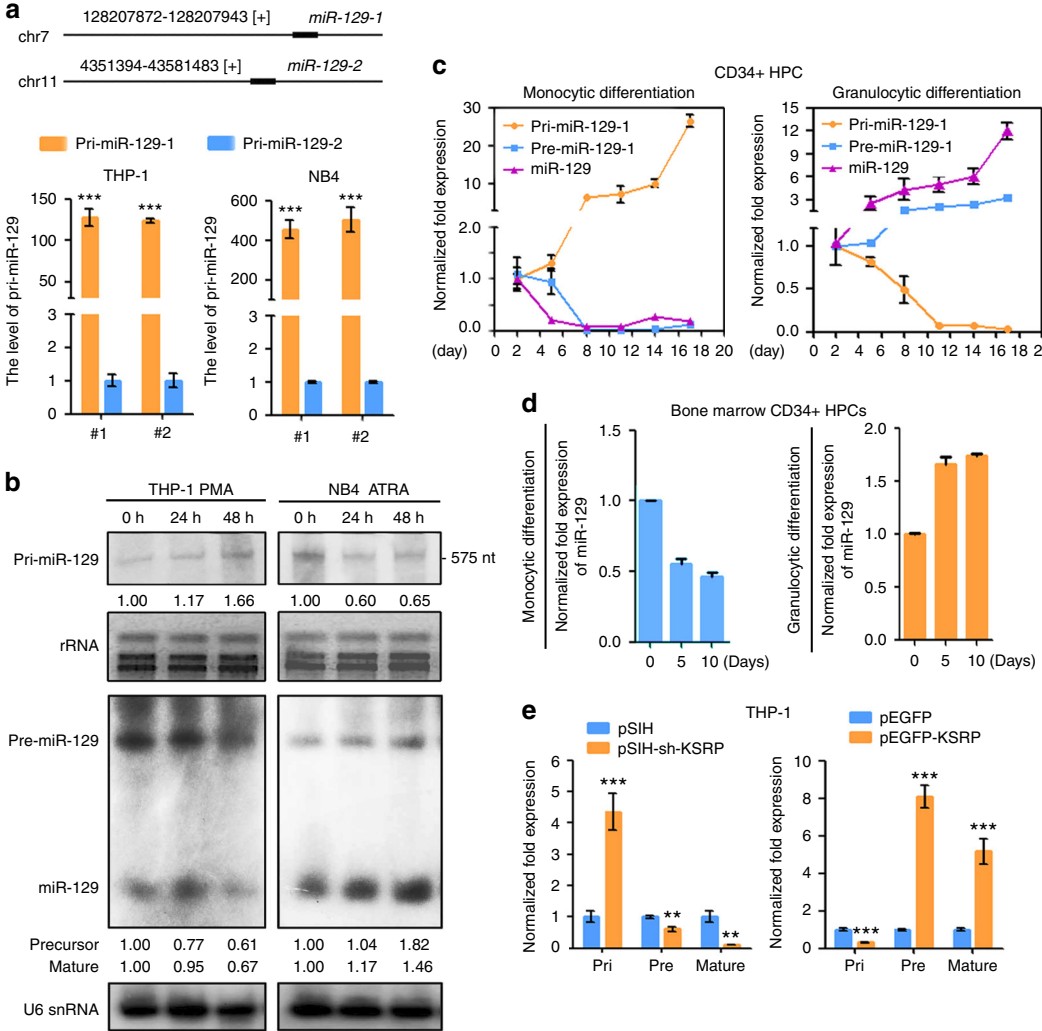

**Fig. 4** KSRP regulates primary miRNA processing in myeloid cells. **a** Upper: schematic representation of two miR-129 loci in the human genome; lower: qPCR analysis of the relative abundance of the two pri-129 transcripts in THP-1 and NB4 cells. The expression of each transcript was normalized to pri-129-2 using the ΔΔCq method. Three technical replicates from a single experiment representative of two independent experiments. **b** Northern blot analysis of miR-129 transcripts in THP-1 cells during monocyte differentiation and NB4 cells during granulocyte differentiation. The upper panels show the hybridization of pri-129-1 with digoxigenin (DIG)-labeled RNA probes; rRNA was used as a loading control. The lower panels show the hybridization of pre-miR-129-1 and miR-129 with isotope-labeled DNA probes; U6 snRNA was used as a loading control. Relative expression of pri-, pre-, and mature miR-129 was quantified using Image J software. **c** qPCR analysis of the expression of pri-129-1, pre-miR-129-1 and miR-129 in the CD34$^+$ HPC models of monocytic and granulocytic differentiation. **d** qPCR analysis of the expression of miR-129 in the monocytic and granulocytic differentiation of bone marrow derived CD34$^+$ HPCs. **e** Left: qPCR of the expression of pri-129-1, pre-miR-129-1 and miR-129 in THP-1 cells transfected with pSIH-sh-KSRP or pSIH control. Right: qPCR of the expression of pri-129-1, pre-miR-129-1 and miR-129 in THP-1 cells transfected with pEGFP-KSRP or the control. Three technical replicates from a single experiment. Data are shown as means±s.d. **$P < 0.01$, ***$P < 0.001$, Student's $t$-test

subset of miRNAs[25], we investigated whether KSRP promotes pri-129-1 processing through a similar mechanism (Fig. 6a). The in vivo interaction between KSRP and the Drosha-DGCR8 complex in THP-1 cells was examined and confirmed to occur in an RNA-independent manner using a co-immunoprecipitation (Co-IP) assay (Fig. 6b). Moreover, RIP-RT-PCR analysis performed in 293T showed that pri-129-1 and pri-let7a were less precipitated upon KSRP knockdown, whereas pri-23b showed no obvious changes (Fig. 6c), suggesting that the presence of KSRP specifically facilitated the binding of DGCR8 and Drosha to its targeted pri-miRNAs. To eliminate the possibility that their interaction was an artifact arising during cell lysis[26], an in vivo in situ proximity ligation assay (PLA) was performed (Fig. 6d). Primary antibodies against KSRP and Drosha or DGCR8 were added. A combination of anti-KSRP antibody and normal anti-

rabbit IgG was used as negative controls. Considerable number of proximity signals (dots) per nucleus was detected in samples incubated with anti-KSRP and anti-Drosha or anti-KSRP and anti-DGCR8 antibodies (Fig. 6e; Supplementary Fig. 5c), but not in the negative ones.

Transcribed wild-type (129_WT) or KSRP-binding site mutant (129_Mut) pri-129-1 substrates were subjected to processing by HeLa cell nuclear extracts (NEs), which possess abundant amounts of the Drosha-DGCR8 complex, to determine whether KSRP modulates pri-129-1 processing in vitro. The addition of increasing amounts of the KSRP protein enhanced the cleavage efficiency of 129_WT, whereas KSRP only slightly affected 129_MUT processing (Fig. 6f). Consistently, pri-let-7a processing, but not pri-23a processing, was also facilitated by increasing amounts of KSRP (Supplementary Fig. 5a, b); the addition of

bovine serum albumin (BSA) did not change the processing of all the transcripts. Additionally, the re-introduction of KSRP proteins could rescue the deceleration in pri-129-1 processing caused by KSRP depletion in NEs treated with an anti-KSRP antibody (Fig. 6g, h). Therefore, the binding of KSRP and subsequent recruitment of the Drosha-DGCR8 complex promotes pri-129-1 processing.

**KSRP promotes pri-129-1 processing in vivo**. GFP-labeled wild-type (129_WT) or KSRP-binding site mutant pri-129 constructs

(129_Mut1~4 and 129_MutF) (Fig. 7a) were co-transfected with the pCMV6-KSRP construct into 293T cells to verify the above in vivo findings. A RFP plasmid was co-transfected simultaneously to calculate the transfection efficiency. KSRP markedly reduced the fluorescence of 129_WT and 129_Mut3, but mutations in the other three sites of pri-129 (129_Mut1, 129_Mut2 and 129_Mut4) alleviated this effect (Fig. 7b, c). In contrast, the fluorescence of 129_MutF, which contains all four mutations, was unaffected by KSRP overexpression (Fig. 7b, c). Thus, KSRP promotes pri-129-1 cleavage mainly by binding to sites 1, 2, and 4.

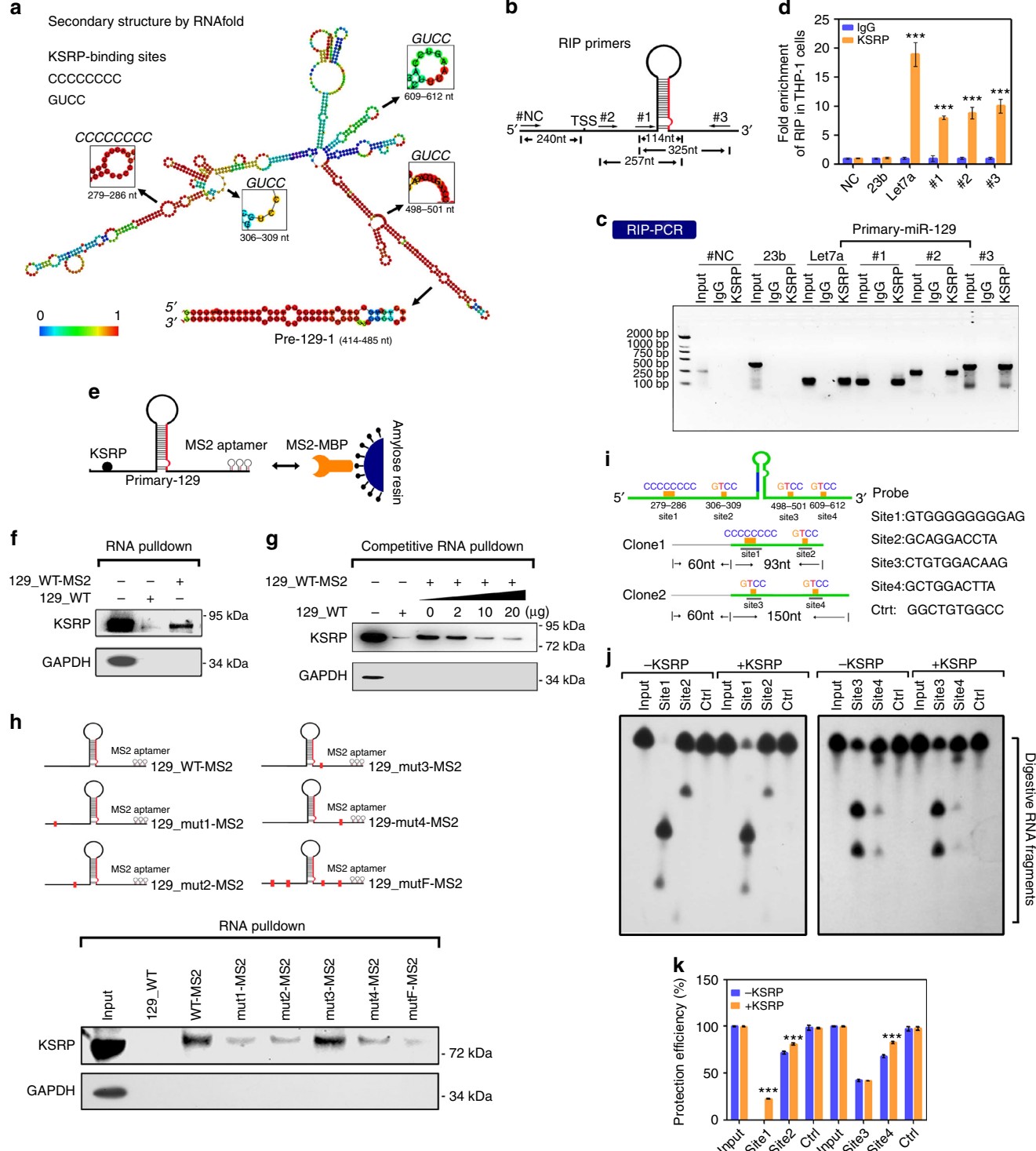

　　　　

Finally, the regulation of pri-129-1 processing by KSRP was investigated during the development of zebrafish embryos. Transcribed GFP-pri-129-1 fusion mRNAs and the KSRP mRNA were microinjected into zebrafish zygotes (Fig. 7d), and GFP alone was injected with the KSRP mRNA as a control. As expected, KSRP decreased the fluorescence of 129_WT in 4 h.p.f. and 24 h.p.f. zebrafish embryos after the microinjection (Fig. 7e, f; and Supplementary Fig. 5d, e). However, the fluorescence of the 129_MutF and GFP control was similar with or without KSRP expression (Fig. 7e; and Supplementary Fig. 5d, e).

**miR-129 regulates myeloid differentiation**. The association of KSRP and pri-129-1 processing prompted us to question whether miR-129 also has important roles in myeloid differentiation. HPCs were transduced with lentiviruses expressing miR-129 (lenti-129), miR-129 inhibitors (miRZIP-129) or their controls and then induced to differentiate into monocytes or granulocytes to examine the role of miR-129 in myeloid differentiation. In both gain- and loss-of-function analyses, miR-129 inhibited monocyte differentiation but promoted granulocyte differentiation, as indicated by changes in the percentages of $CD14^+/CD11b^+$ or $CD11b^+$ cells (Fig. 8a, b; and Supplementary Fig. 6a, b), cell morphological maturation (Fig. 8c; and Supplementary Fig. 6a, b), colony formation (Fig. 8d, e) and *CD14* or *CD11b* mRNA expression (Fig. 8f, g). Meanwhile, functional analyses were also performed in THP-1 and NB4 cells, which were first transfected with the miR-129 mimic (miR-129) or control mimic (NC) and then stimulated with PMA and ATRA (Supplementary Fig. 7a, e), respectively. miR-129 transfection decreased the percentage of $CD14^+$ cells, *CD14* mRNA expression and the proportion of mature monocytes after PMA treatment (Supplementary Fig. 7a–d). In contrast, miR-129 transfection increased the percentage of $CD11b^+$ cells, *CD11b* mRNA expression and the proportion of mature granulocytes after ATRA treatment (Supplementary Fig. 7e–h).

**miR-129 acts downstream of KSRP**. To illustrate whether KSRP modulates miR-129 production to impact M and G differentiation, a rescue assay was performed by transfecting THP-1 cells with a combination of KSRP (or control) and miR-129 inhibitor (or control) before monocyte differentiation. Co-transfection of KSRP and a miR-129 inhibitor (the "rescue" group) attenuated the decrease in the percentage of $CD14^+$ cells induced by KSRP (Supplementary Fig. 7i, j). In another experiment, the rescue assay in granulocytic NB4 cells used the combination of si-KSRP (or control) and miR-129 mimic (or mimic control) transfection. Accordingly, co-transfection rescued the decrease in the percentage of $CD11b^+$ cells induced by the si-KSRP transfection (Supplementary Fig. 7k, l). Collectively, miR-129, whose processing

was regulated by KSRP, functioned downstream of KSRP to regulate myeloid differentiation.

**miR-129 targets RUNX1 in myeloid cells**. Next we studied the mechanism by which miR-129 differentially regulates myeloid differentiation. RUNX1 was predicted to be a potential target of miR-129 by bioinformatics prediction. Reporter assays revealed that miR-129 reduced the luciferase activities of the positive control (PC), WT and RUNX1_Mut1 reporters compared to their activities in a scrambled control (Fig. 9b). In contrast, the luciferase activities of the RUNX1_Mut2 and RUNX1_Mut3 reporters were not repressed by miR-129, indicating that the repression was dependent on miR-129 binding to sequences at position 1851-1857 and 2459-2467 (Fig. 9a). In contrast to the expression of miR-129, RUNX1 was up-regulated during monocytic induction and downregulated during granulocytic induction in both HPCs and myeloid cell lines (Fig. 9c, Supplementary Fig. 7m). Additionally, the expression of the *RUNX1* mRNA and protein expression was repressed by miR-129 overexpression and increased by miR-129 knockdown (Fig. 9d, e; Supplementary Fig. 7n, o). Therefore, RUNX1 is a direct target of miR-129 in myeloid cells.

To investigate whether RUNX1 targeting is involved in miR-129-mediated myeloid differentiation, HPCs were transduced with lentivirus expressing GFP (lenti-GFP) or RUNX1 (lenti-RUNX1) and then underwent monocytic or granulocytic differentiation. RUNX1 promoted monocyte differentiation, but repressed granulocyte differentiation, as indicated by the changes in cell morphology (Fig. 9f, Supplementary Fig. 6c, d), the percentages of $CD14^+$ or $CD11b^+$ cells (Fig. 9g), and *CD14* (Fig. 9h) or *CD11b* mRNA expression (Fig. 9h).

Next, rescue assays were performed in THP-1 and NB4 cells by transfection with combination of si_RUNX1 (or control) and miR-129 inhibitor (or control), in which si_RUNX1 restored the increase of endogenous RUNX1 protein induced by miR-129 inhibitor (Supplementary Fig. 8a, c). After PMA or ATRA induction, co-transfection rescued miR-129 inhibitor–stimulated CD14 increase as well as CD11b reduction (Supplementary Fig. 8b, c, e, f), indicating that miR-129-mediated RUNX1 repression directs the divergent differentiation of monocytes and granulocytes. We noticed that CD11b positive cells in NB4 cells were almost unchanged between two different transfection methods with differing efficiencies (Supplementary Fig. 8). It is possible that off-target effect of si-RUNX1 may exist, but it is more likely that granulocytic differentiation of NB4 cells is too sensitive to cover the modulation of RUNX1 protein level.

**miR-129 controls myeloid differentiation via M- or G-CSFR**. RUNX1 is a member of the core binding factor (CBF) family of proteins[27]. In previous studies, RUNX1 promotes monocytic

**Fig. 5** KSRP binds to pri-129-1 transcripts in myeloid cells. **a** RNAfold software was used to predict the secondary structure of the pri-129-1 transcripts, and the four KSRP-binding sites are highlighted. **b** Schematic representation of the three pairs of primers used in the RIP assay. **c** Agarose gel electrophoresis showing the results of the RIP assay in THP-1 cells. **d** RIP-qPCR analysis of the co-precipitation of the pri-129-1, pri-23b and pri-let7a transcripts by the KSRP antibody compared to the IgG control. Three technical replicates from a single experiment. **e** Schematic representation of an RNA pull-down assay using MS2-tagged pri-129-1 affinity purification. **f** Immunoblot of endogenous KSRP in RNA pull-down assays from 293 T cells transfected with either MS2-tagged wild-type pri-129-1 (129_WT-MS2) or non-MS2-tagged pri-129-1 (129_WT). An unrelated protein (GAPDH) was used as the control. **g** Immunoblot of endogenous KSRP in competitive RNA pull-down assays from 293T cells co-transfected with 129_WT-MS2 and increasing doses of 129_WT. **h** Upper: schematic representation of the construction of the KSRP-binding site mutant plasmids; lower: immunoblot of endogenous KSRP in the RNA pull-down assay from 293 T cells transfected with either the MS2-tagged wild-type pri-129-1 (129_WT-MS2) or the MS2-tagged KSRP-binding site mutant (129_mut1-MS2, 129_mut2-MS2, 129_mut3-MS2, 129_mut4-MS2 and 129_mutF-MS2) constructs. **i** Schematic representation of the RNA transcripts and DNA probe sequences used in the RNase H protection assay. **j** RNase H protection assay using probes in **i**. Size of markers for both clones are the same. **k** Quantitation of the KSRP protection efficiency using Image Pro, as indicated in **j**. Three technical replicates from a single experiment. Data are shown as means±s.d. ***$P < 0.001$, Student's *t*-test

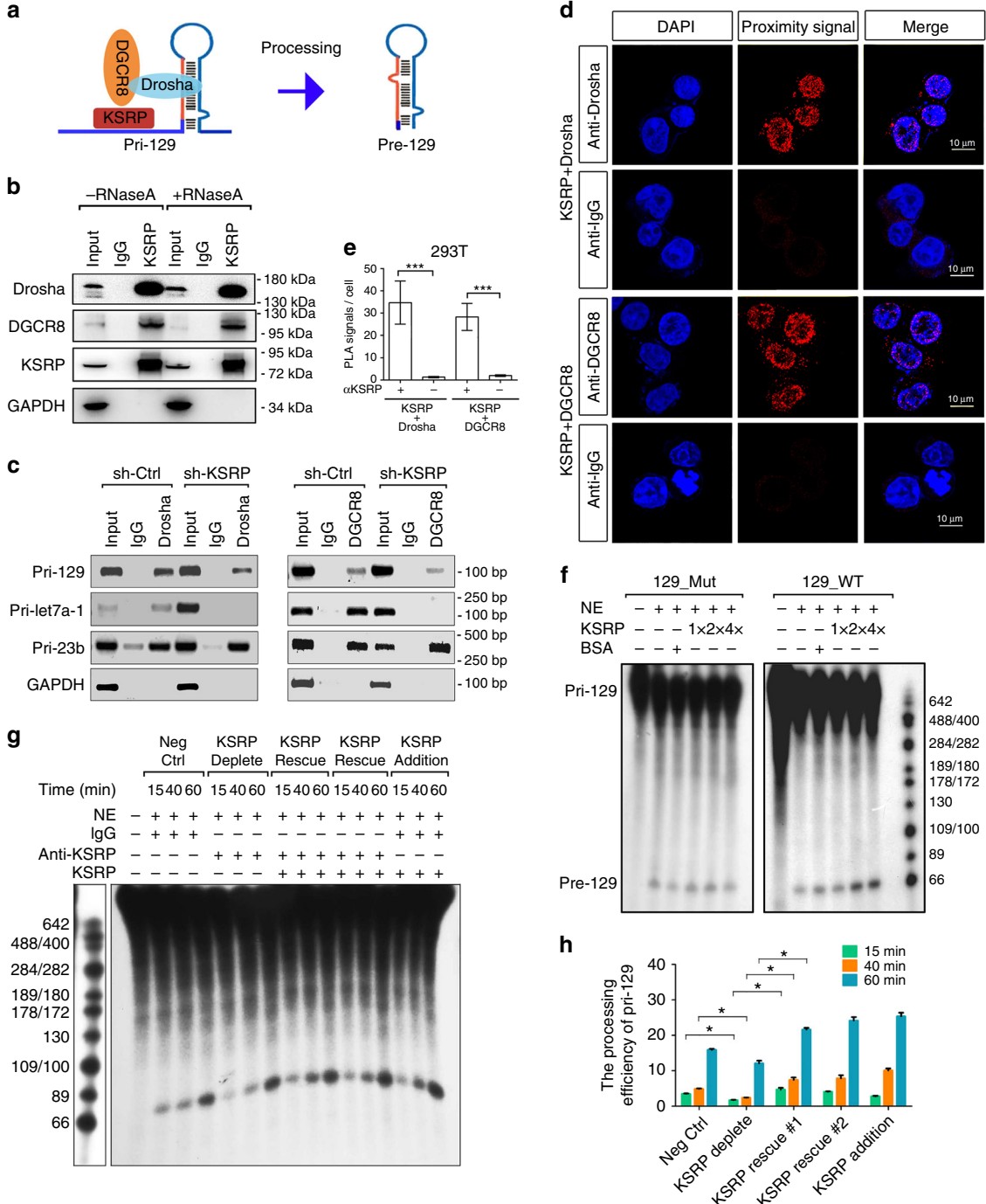

**Fig. 6** KSRP promotes the processing of pri-129-1 in vitro. **a** Schematic representation of the involvement of KSRP in the pri-129-1 processing model. **b** As shown in the co-IP analysis, KSRP interacts with the Drosha-DGCR8 complex in THP-1 cells in an RNA-independent manner. This experiment was performed by incubating the KSRP immunoprecipitates with or without RNAse A (0.1 mg/mL) at 37 °C for 20 min. **c** Agarose gel electrophoresis showing the results of the RIP-PCR of pri-129-1, pri-23b and pri-let7a was performed by immunoprecipitating the complexes in KSRP knockdown or control 293T cells with Drosha and DGCR8 antibodies. **d** Representative images for in situ proximity ligation assay (PLA) of KSRP with Drosha or DGCR8 in 293T cells (Scale bar, 10 μm; Original magnification, 630×). **e** Quantification of PLA results (n ≥ 3; cell numbers > 100 per experiment) were shown. **f** In vitro pri-129-1 processing reactions used isotope-labeled wild-type pri-129-1 (129_WT) or pri-129-1 with KSRP-binding site mutants (all four sites mutant, 129_Mut) as the substrate and were then pre-incubated with HeLa NEs and various amounts of purified KSRP protein, as described in the Methods. **g** Rescue of the KSRP levels in NEs after KSRP depletion with an anti-KSRP antibody further promotes pri-129-1 processing in the in vitro processing assay, as indicated in **f**. **h** Quantitation of the in vitro processing assay shown in **g**. Similar significant differences (p-value < 0.05, two-tailed Student's t-test) were obtained between the KSRP depletion group and the KSRP rescue #1 or #2 groups. Three technical replicates from a single experiment. Data are shown as means±s.d. ***P < 0.001, Student's t-test

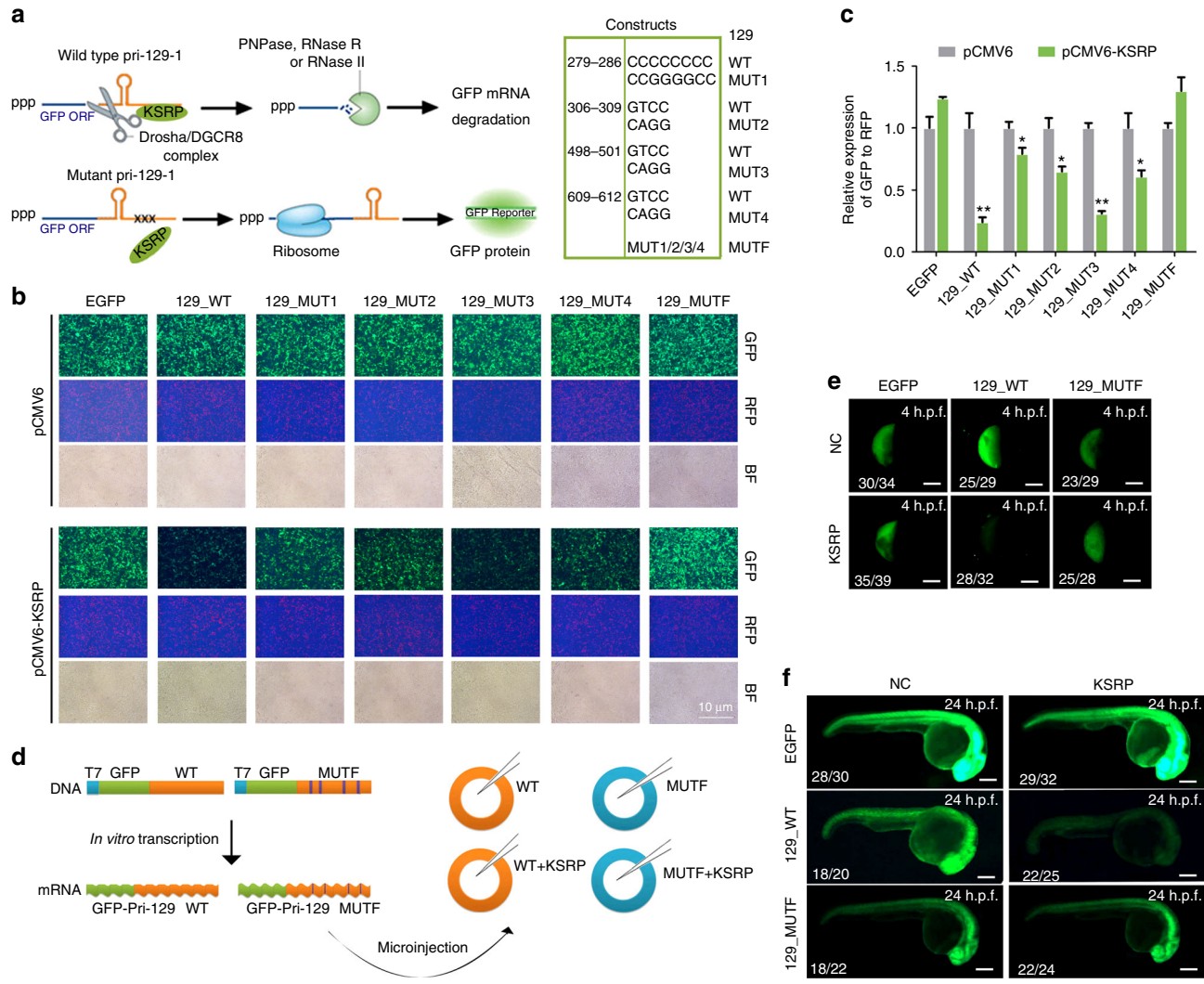

**Fig. 7** KSRP promotes the processing of pri-129-1 in vivo. **a** Schematic representation of the in vivo processing assay performed by co-transfecting GFP-labeled pri-129-1 (129_WT) or GFP-labeled KSRP-binding site mutant (129_Mut1, 2, 3, 4 for each site and 129_MutF for all four sites) constructs with the KSRP construct or empty vector. The mutant sequences in the constructs are shown in the right panel. **b** Images of GFP and RFP fluorescence in 293 T cells used in the in vivo processing assay, as indicated in h (Scale bar, 100 μm). RFP fluorescence from a co-transfected plasmid was used as control for transfection efficiency. **c** The quantitative ratio of GFP expression to RFP expression in **b**. **d** Schematic representation of the in vivo processing assay in zebrafish, as described in the Methods. **e**, **f** Images of GFP fluorescence in the 4 h.p.f. **e** or 24 h.p.f. **f** zebrafish embryos at after the zebrafish zygote was microinjected with a combination of GFP-pri-129_WT or GFP-pri-129_MutF fusion mRNAs with the KSRP mRNA or the control (Scale bar, 200 μm). Data are shown as means±s.d. *P < 0.05, **P < 0.01, Student's t-test

differentiation by transactivating the M-CSFR promoter, whereas it inhibited granulocyte differentiation by repressing the G-CSFR promoter[28,29]. Accordingly, RUNX1-overexpressing HPCs expressed higher levels of the *M-CSFR* mRNA during monocyte differentiation but lower levels of the *G-CSFR* mRNA during granulocyte differentiation compared to the lenti-GFP-transduced cells (Fig. 9h). Similar results were obtained when miR-129 expression was inhibited in HPCs (Fig. 8g). However, miR-129 overexpression resulted in lower M-CSFR but higher G-CSFR expression compared with control (Fig. 8f). Thus, these results extended the hypothesis that miR-129 targets RUNX1 to determine the myeloid differentiation outcome by adding M-CSFR and G-CSFR to this regulatory network.

**KSRP and miR-129 have consistent roles in vivo**. A human HPC-transplanted mouse model (Fig. 10a) was adopted to confirm the roles of KSRP and miR-129 in vivo. Four weeks after the

transplantation of lenti-GFP- or lenti-sh-KSRP-transduced CD34+ HPCs, 19%–53% of the bone marrow (BM) cells were CD33+ (Supplementary Fig. 9a), compared with < 1% before transplantation, indicating the successful introduction of human HPCs. KSRP silencing increased CD14 expression and decreased CD11b expression in BM myeloid cells (Fig. 10b, Supplementary Fig. 9b). Moreover, mice transplanted with lenti-129- or lenti-GFP-transduced-HPCs were also used for a myeloid differentiation analysis (Fig. 10c). As expected, miR-129 over-expression decreased the percentage of CD14+ cells and increased the percentage of CD11b+ cells (Fig. 10d, Supplementary Fig. 9c). These results, combined with the ex vivo data, indicated the consistent inhibitory roles of KSRP and miR-129 in monocyte differentiation and stimulatory roles in granulocyte differentiation.

**The KSRP/miR-129/RUNX1 axis in myeloid differentiation**. Based on the above findings, we postulated that KSRP, miR-129

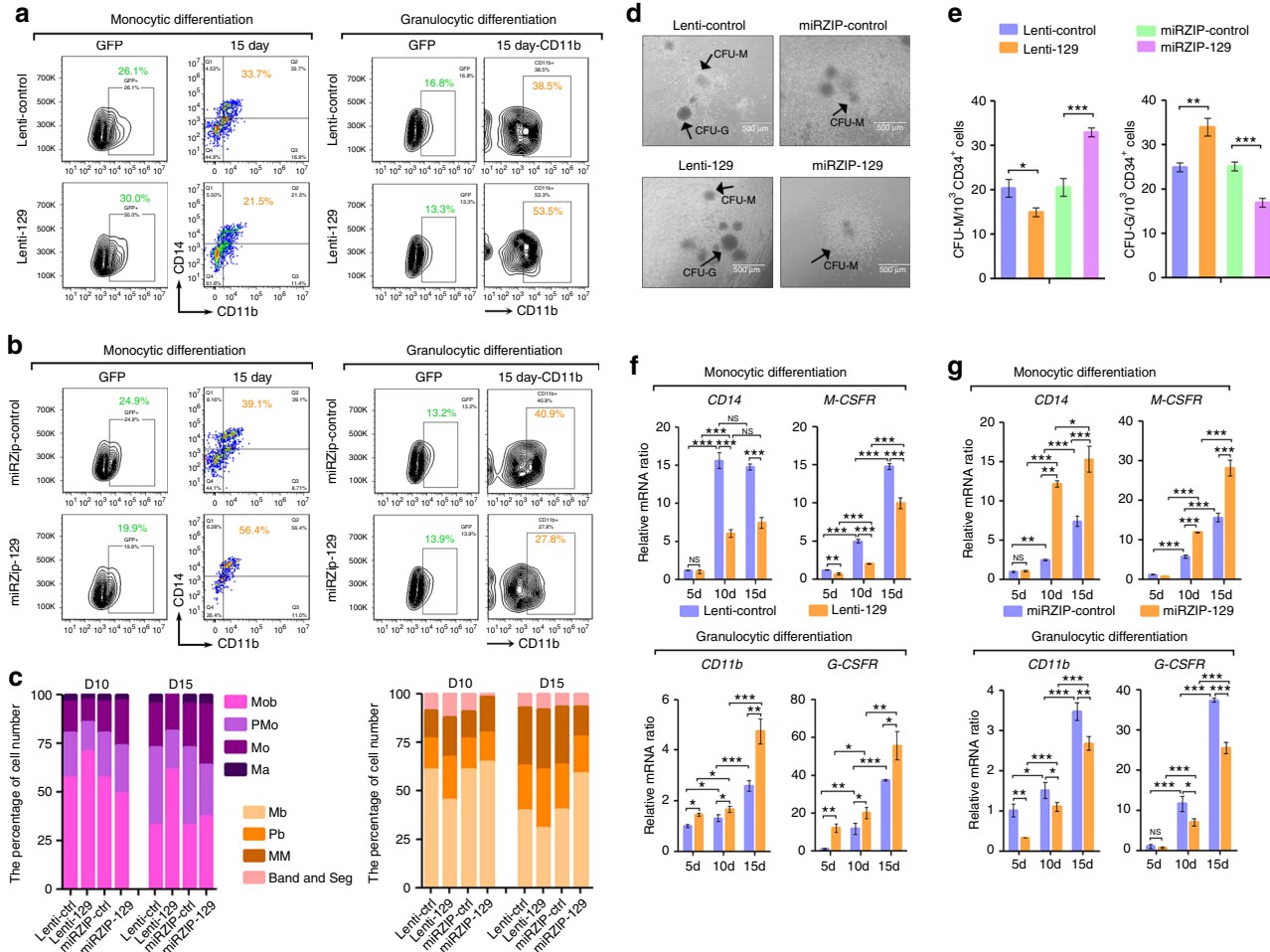

**Fig. 8** miR-129 regulates monocytic and granulocytic differentiation. **a**, **b** CD34[+] HPCs were transduced with lenti-129 or lenti-control **a** or miRZIP-129 or miRZIP-control **b** for 24 h, and then cultured for 15 days to allow cells to differentiate into monocytes or granulocytes. FACS analysis of the CD14[+]/CD11b[+] and CD11b[+] populations of GFP-positive cells on day 15. **c** Quantification of Giemsa staining. **d** Colony-forming assay of CD34[+] HPCs transduced with lenti-129, miRZIP-129 or their controls. The bar represents 100 μm. **e** The statistical analysis of the colony numbers shown in **d**. Three technical replicates from a single experiment. **f**, **g** qPCR analysis of the monocyte-granulocyte markers at the indicated times after miR-129 overexpression **f** or miR-129 knockdown **g** in CD34[+] HPCs undergoing monocytic (upper) and granulocytic (lower) differentiation. Three technical replicates from a single experiment. Data are shown as means±s.d. *P < 0.05, **P < 0.01, ***P < 0.001, Student's t-test

and RUNX1 participate in a regulatory axis to regulate the outcome of myeloid differentiation (Fig. 10e). BM from transplanted mice (lenti-GFP, lenti-sh-KSRP or lenti-129) was analyzed to test this hypothesis. Indeed, KSRP knockdown increased pri-129 levels and decreased mature miR-129 levels (Fig. 10f). Meanwhile, miR-129 overexpression suppressed RUNX1 expression in 3 out of 4 mice (Fig. 10g). Consequently, KSRP knockdown further increased RUNX1 expression in BM cells (Fig. 10h). Therefore, the divergent regulatory activity of KSRP was dependent on miR-129 and its direct repression of RUNX1 expression during myeloid differentiation. To define the reach of our findings under physiological condition, KSRP and miR-129 expression was detected in monocytes and granulocytes from normal peripheral blood (Supplementary Fig. 10a). As expected, both KSRP and miR-129 levels were higher in neutrophils compared with that in monocytes in 9 of the 11 subject (Supplementary Fig. 10b, c). Overall, our result indicates that KSRP and miR-129 concomitantly expressed in mature blood cells, and their high levels tend to favor granulocyte maturation while lower expression favors monocyte maturation.

## Discussion

Myeloid differentiation is a process that replaces and/or increases the number of cells that function in nonspecific defense and help initiate specific defense mechanisms[28]. This process is orchestrated by both transcriptional and post-transcriptional mechanisms that are highly responsive to environmental stimuli[30]. In this study, we identified KSRP as a specific regulator that promotes granulocytic and inhibits monocytic differentiation, and its function relies on the directly enforced processing of pri-129-1 and indirect attenuation of RUNX1.

The introduction of large-scale quantitative screens has provided methods for the investigation of post-transcriptional mechanisms at a system biology level. In this study, the transcriptome of HPCs was characterized during myeloid differentiation, and the DEGs were determined to characterize the major RBPs. The present work revealed the dynamic expression and essential function of several candidate RBPs that may control myeloid differentiation, providing a complete picture of RBPs that affect post-transcriptional regulatory networks involved in this process. However, these RBPs may regulate myeloid

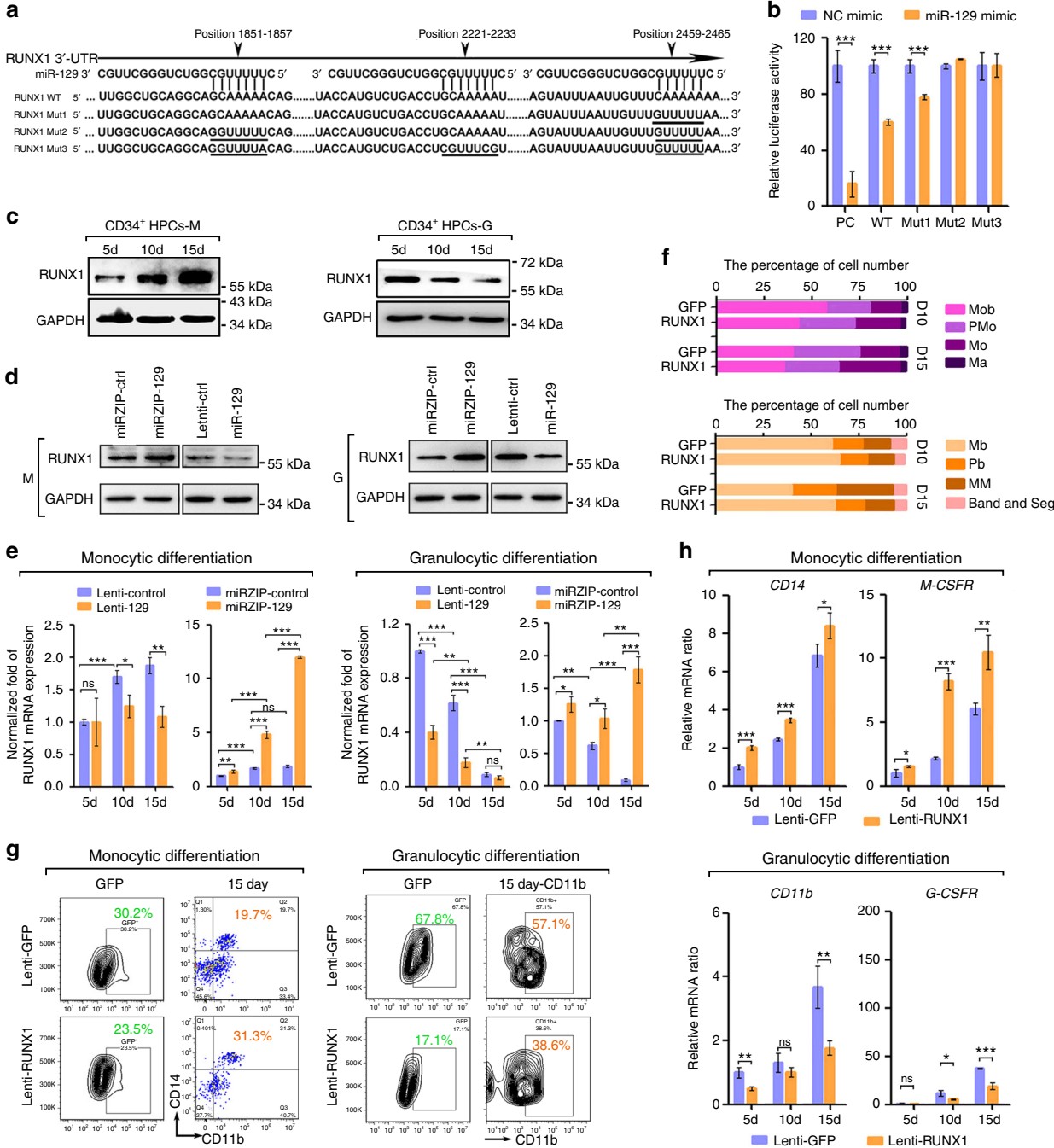

**Fig. 9** miR-129 targets RUNX1 to modulate myeloid differentiation. **a** A computational prediction of the conserved binding sites and the designed mutated sequences in these sites within the 3′ UTR of the human RUNX1 mRNA. **b** Relative luciferase activity of the indicated reporter constructs. PC, positive control. Three technical replicates from a single experiment representative of two independent experiments. **c** Immunoblot of the endogenous RUNX1 levels in CD34[+] HPCs undergoing monocytic or granulocytic differentiation. **d** Immunoblot of RUNX1 expression in CD34[+] HPCs transduced with lenti-129, miRZip-129 or their controls on day 10 of monocytic or granulocytic differentiation. **e** qPCR of the *RUNX1* mRNA levels in CD34[+] HPCs transduced with lenti-129, miRZip-129 or their controls during monocytic and granulocytic differentiation. Three technical replicates from a single experiment. **f**, **g** CD34[+] HPCs were transduced with lenti-RUNX1 or lenti-GFP control for 24 h and then cultured for 15 days to allow the cells to differentiate into monocytes or granulocytes. Quantitation of the Giemsa staining was shown in **f** and FACS analysis of the CD14[+]/CD11b[+] and CD11b[+] populations of GFP-positive cells on day 15 (**g**). **h** qPCR analysis of the monocyte-granulocyte markers at the indicated times during monocytic or granulocytic differentiation of RUNX1-overexpressing cells. Three technical replicates from a single experiment. Data are shown as means±s.d. *$P < 0.05$, **$P < 0.01$, ***$P < 0.001$, Student's *t*-test

differentiation through diverse pathways. Importantly, the application of qPCR following RNA-seq revealed a divergent expression pattern of KSRP during G-CSF- or M-CSF-induced differentiation of HPCs. Because the data regarding the expression of these RBPs was obtained in CD34[+] cells, we cannot conclude that changes in RBP coverage represent the changes

occurring in vivo. Future validation in animals and the development of reliable animal models are necessary to characterize the functions of RBPs.

As an AU-rich RNA-binding protein (AUBP), KSRP interacts with single-stranded adenylate/uridylate-rich elements (ARE)-containing mRNAs and was originally shown to mediate

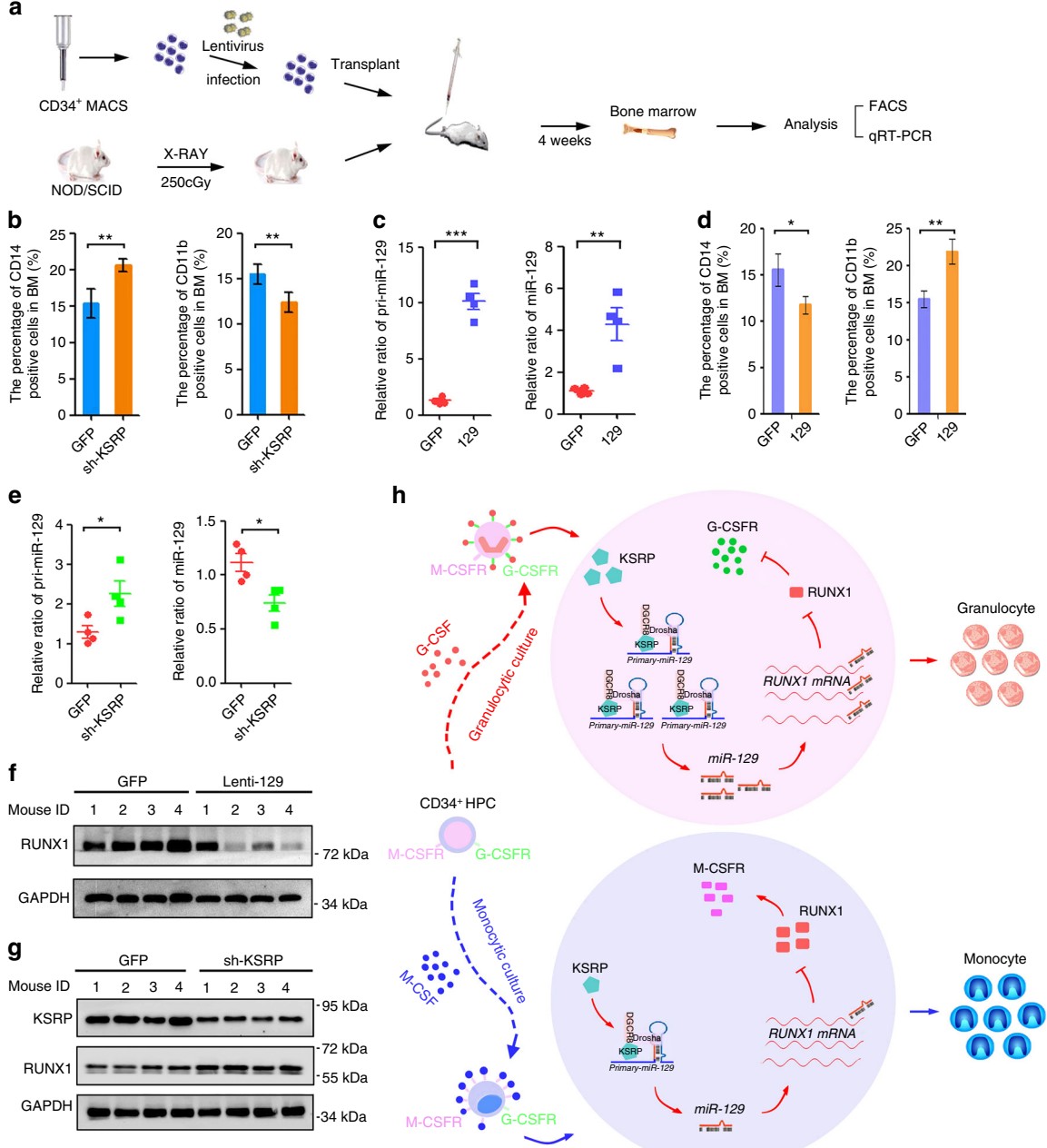

**Fig. 10** KSRP and miR-129 regulate monocytic and granulocytic differentiation in vivo. **a** A schematic representation of the in vivo experimental design for the transplantation of CD34+ HPCs isolated from cord blood into mice (n = 4). **b** The CD14+ and CD11b+ populations in BM from mice transplanted with lenti-sh-KSRP or lenti-GFP-transduced HPCs were monitored by FACS. **c** qPCR analysis of pri-miR-129 and mature miR-129 expression in BM from mice transplanted with the lenti-GFP- or lenti-129-transduced HPCs. **d** The CD14+ and CD11b+ populations in BM from mice transplanted with lenti-129- or lenti-GFP-transduced HPCs were monitored by FACS. The quantitation is shown in the left panel. **e** Schematic representation of a model describing the role of the KSRP-miR-129-RUNX1 axis in regulating the normal lineage-specific differentiation of CD34+ HPCs. **f** qPCR analysis of pri-miR-129 and mature miR-129 expression in BM from mice transplanted with lenti-GFP- or lenti-sh-KSRP-transduced CD34 + HPCs. **g** Immunoblot of RUNX1 expression in BM from mice transplanted with lenti-129- or lenti-GFP-transduced HPCs. **h** Immunoblot of KSRP and RUNX1 expression in BM from mice transplanted with lenti-sh-KSRP- or lenti-GFP-transduced HPCs. Data are shown as means±s.d. *P < 0.05, **P < 0.01, ***P < 0.001, Student's t-test

mRNA splicing[31] and mRNA decay[32,33]. Despite its function in innate immune cell responses[34,35] and mesenchymal cell differentiation[36], we first identified and characterized the role of KSRP in regulating canonical myeloid differentiation of CD34+ hematopoietic stem cells. Conversely, dysregulated KSRP function may result in differentiation arrest and certain types of myeloid malignancies.

KSRP promotes miRNA maturation at both the precursor and primary transcript level. KSRP serves as an integral component of both Drosha and Dicer complexes and regulates the biogenesis of a subset of miRNAs by binding to the terminal loop of the target miRNA precursors, which is required for specific biological programs, including proliferation, apoptosis, differentiation and inflammation[25,37,38]. In this study, one mechanism by which

KSRP regulates myeloid differentiation is to facilitate pri-miR-129 processing. However, in contrast to the reported interaction that is limited to pri-miRNA stem-loop structures, three G-rich sequences far from pri-129-1 stem-loops bound to KSRP, which then promoted pri-129-1 processing by recruiting the Drosha-DGCR8 complex to this pri-miRNA. The molecular interactions among KSRP, pri-129-1 and Microprocessor at both the molecular and structural levels are waiting to be addressed in the future with the progress of technology. Recently, it was found that in addition to secondary structures, primary sequence features of pri-miRNA are required for efficient processing[39]. Our study also shows that KSRP could bind to sites in the primary sequence of the pri-miRNA to recruit Microprocessor, indicating that sequences other than stem-loops should be considered in the context of RBP binding.

The integration of networks governed by post-transcriptional effectors such as miRNAs, RBPs and transcriptional factors jointly regulates hematopoiesis, with important implications for development and disease[40]. A series of rescue experiments demonstrated that KSRP, miR-129 and RUNX1 form an axis to regulate differentiation and monocytic/granulocytic lineage commitment (Fig. 10h). Increased miR-129 processing by KSRP is definitely one mechanism underlying this regulatory pathway, but interactions with other components may extend this axis, and the balance between positive and negative regulators may fine-tune the efficiency of miR-129 processing and protein expression. For example, RBP itself is subject to miRNA-mediated regulation through translational repression by several miRNAs[34,41]. In addition, RUNX1 tends to affect the amount of KSRP and miR-129 at the transcriptional level, forming a cross-regulatory loop.

As a master transcriptional regulator, RUNX1 expression is essential for the establishment of definitive hematopoiesis of all lineages, because RUNX1 knockout mice display embryonic lethality[42]. The upregulated expression pattern and role of Runx1 in promoting monocyte differentiation has been unanimously elucidated[42,43], and this mechanism relies on the transactivation of M-CSFR[28,43]. In addition, RUNX1 was reported to suppress G-CSFR expression, leading to restricted granulopoiesis[29,44]. The binding and activation of the Pu.1 distal enhancer by RUNX1 also inhibits granulocytic commitment, as revealed in knockout mice[45]. However, decreased RUNX1 activity due to mutations or haploinsufficiency was also reported to result in arrest of granulocytic maturation[46–48]. We speculate that RUNX1 control of granulopoiesis may depend on its origin and its tropism for binding downstream regulators. Therefore, the multifactorial nature of the regulatory networks should be addressed, where a fine modulation of one component may have a substantial effect on another component and subsequently perturb the balance.

Besides the processing of pri-miR-129, we still do not know whether other processes such as mRNA decay or maturation of other miRNAs[49,50], the two most acknowledged mechanisms, are also involved in the divergent regulation of KSRP during myeloid differentiation. It is known that KSRP regulates immune responses through regulating multiple inherently instable mRNAs coding for pro-inflammatory mediator by ways of mRNA decay or translational inhibition, as well as promoting maturation of a set of miRNAs[35,36]. On the basis of these evidences, we can suppose that KSRP could have complex functions in myeloid differentiation. Overall, the diversity in KSRP target selection may depend on cell types, physiological conditions and are still largely unknown due to technical limitations.

Taken together, our results illustrate the role of the KSRP/miR-129/RUNX1 axis in myeloid lineage differentiation. Regulation by KSRP appears similar to miRNA-mediated post-transcriptional regulation in that individual miRNAs also bind to multiple target mRNAs and down-regulate their expression levels, often by only

20–50%[51]. A comprehensive understanding of the RBP–ncRNA interaction will help researchers discover mechanisms that contribute to normal development and complex genetic diseases at post-transcriptional level, which should be an important aspect of future investigations.

## Methods

**Monocytic and granulocytic differentiation of cell lines**. The human monocytic cell line THP-1, and the promyelocytic cell line NB4 and HL-60 were purchased from ATCC (Manassas, USA) and maintained in RPMI1640 supplemented with 10% fetal bovine serum (Gibco, Carlsbad, CA). Cells were screened by PCR for authentication and by immunofluorescence tests for freedom from mycoplasma contamination. Monocytic differentiation of THP-1 and HL-60 cells was induced by treating the cells with 50 nM PMA (Sigma-Aldrich, Deisenhofen, Germany) for 0, 24, 48, and 72 h. Granulocytic differentiation of NB4 and HL-60 cells was induced by treating the cells with 1 μM ATRA (Sigma-Aldrich, Deisenhofen, Germany). No cell lines used in this study were found in the database of commonly misidentified cell lines that is maintained by ICLAC and NCBI Biosample. Cell lines were tested for mycoplasma contamination but have not been re-authenticated.

**Sorting CD34+ HPCs**. Human umbilical cord blood (UCB) was obtained from normal full-term deliveries at Peking Union Hospital, and bone marrow (BM) from normal donors were obtained from the 307th Hospital of Chinese People's Liberation Army. Informed consent and approval of the Research Ethics Committee was obtained. Mononuclear cell (MNC) fractions were isolated from UCB and BM using a Percoll density gradient (d = 1.077; Amersham Biotech, Germany). CD34+ cells were enriched from MNCs using positive immunomagnetic selection (CD34+ MultiSort Kit, Miltenyi Biotec, Bergisch-Glad-bach, Germany).

**Monocytic differentiation of HPCs**. The enriched CD34+ hematopoietic progenitor cells (HPCs) were cultured in high glucose IMDM supplemented with 30% fetal bovine serum (Hyclone, Logan, Utah), 1% BSA, 100 μM 2-ME, 2 ng/mL recombinant human IL-3, 100 ng/mL recombinant human SCF, 50 ng/mL recombinant human M-CSF, 100 ng/mL recombinant human FLT3, 1 ng/mL recombinant human IL-6, 60 mg/mL penicillin and 100 mg/mL streptomycin to induce monocyte differentiation. Cytokines were purchased from Peprotech (Rocky Hill, NJ). Cells were harvested every 3–5 days. For morphological analyses, cells were smeared on glass slides, fixed in methanol 10 min, stained with May-Grünwald/Giemsa, and analyzed at 400x magnification under a microscope (Nikon TE2000, Japan) equipped with a digital camera.

**Granulocytic differentiation of HPCs**. The enriched CD34+ hematopoietic progenitor cells (HPCs) were cultured in high glucose IMDM supplemented with 30% fetal bovine serum (Hyclone, Logan, Utah), 1% BSA, 100 μM 2-ME, 2 ng/mL recombinant human IL-3, 100 ng/mL recombinant human SCF, 20 ng/mL recombinant human G-CSF, 10 ng/mL recombinant human IL-6, 60 mg/mL penicillin and 100 mg/mL streptomycin to induce granulocyte differentiation. Cytokines were purchased from Peprotech (Rocky Hill, NJ). Cells were harvested every 3–5 days. Morphological analyses were performed in the same way as monocytic differentiation.

**Oligonucleotides and constructs**. miR-129 mimics (5′-CUUUUUGCGGUCUGG GCUUGC-3′), miRNA inhibitors (anti-129) and negative control molecules (NC) were obtained from Dharmacon (Austin, TX) and cells were transfected with a final concentration of these constructs of 100 nM using DharmFECT1 (Dharmacon, Austin, TX). siRNAs (specifically for KSRP and RUNX1) and control siRNAs (Si-Con) were synthesized by Dharmacon and cells were transfected with 100 nM siRNAs using DharmFECT1.

For KSRP overexpression, the human KSRP (NM_003685.2) cDNA ORF clone (pEGFP-KSRP) was purchased from Addgene (Addgene, MA). For KSRP knockdown, KSRP sh-RNA was cloned into the pSIH-H1 vector downstream of the H1 promoter (lenti-sh_KSRP) or directly purchased in form of lentiviral constructs (TL311984, Origene, Rockville, MD). For pri-129-1 over-expression, a 700-bp wild-type construct was cloned into the pCMV6 vector (129_WT) or into the pMIRNA1 plasmid (lenti-129). The mutant KSRP-binding sites in pri-129-1 (129_MUT1, 129_MUT2, 129_MUT3, 129_MUT4, and 129_MUTF) were also created in pCMV6 vectors. For miR-129 knockdown, miRZIP-129 and miRZIP-control were purchased from System Biosciences (SBI, Palo Alto, CA). For RUNX1 overexpression, RUNX1 ORF was cloned into the pMIRNA1 vector (Lenti-RUNX1).

For the reporter gene assay of miR-129 targets, the reverse complementary sequence to miR-129 was inserted into the pMIR-reporter downstream of the firefly luciferase gene to generate the positive reporter construct (129_positive). The 3′-UTR of human RUNX1 mRNA was amplified and cloned into the same pMIR-reporter downstream of the firefly luciferase gene to form the wild-type reporter (RUNX1_WT). Mutations of the miR-129 binding sites in the 3′-UTR of human

RUNX1 mRNA sequences were created using the QuickChange Site-directed Mutagenesis kit (RUNX1_Mut1, RUNX1_Mut2, and RUNX1_Mut3). Cells were transfected with these constructs using either Lipofectamine 2000 (Invitrogen, Carlsbad, CA) for 293TN cells or Lipofectamine LTX&PLUS for THP-1 cells and NB4 cells, according to the manufacturer's protocols. All primers are listed in Supplementary Data 7.

**Lentivirus production.** The appropriate lentivirus packaging kit was purchased from System Biosciences (SBI, Palo Alto, CA) and used according to the manufacturer's guidelines. The harvested viral particles (lenti-129, lenti-miRZIP-129, lenti-sh_KSRP, lenti-RUNX1, lenti-sh_RUNX1, and lenti-GFP) were added to CD34[+] HPCs, THP-1 or NB4 cells, and the cells were then washed with IMDM after 24 hours and plated for subsequent experiments.

**miRNA deep sequencing.** THP-1 cells were electroporated with pEGFP-KSRP or pEGFP using the Neon transfection system and then sorted by FACS to collect the GFP-positive cells. Small RNAs were isolated from the total RNA with the mirVana miRNA Isolation Kit (Ambion, Austin, TX), according to the manufacturer's instructions. Small RNA libraries and miRNA sequencing were performed by Genergy Bio (Genergy Bio-technology Co., Ltd., Shanghai, China) using an Illumina HiSeq2000 (Illumina, USA). The sequencing data were filtered, the expression profiles were constructed and the adapter dimer reads were separated from the miRNA reads that had already been identified in miRbase 17.0. RNA-seq data were analyzed by Genergy Bio and a list of miRNAs whose expression was increased upon KSRP overexpression is shown in Supplementary Data 5, 6.

**Poly(A)-enriched RNA deep sequencing.** Total RNA was extracted using Trizol reagent (Invitrogen, Carlsbad, CA), according to the manufacturer's instructions, to prepare the poly(A)-enriched RNA-seq library. Poly(A)-enriched RNA libraries and deep RNA-seq were adapted to Hiseq2000 (Illumina, San Diego, CA) and were performed by Genergy Bio. A complete list of primary miRNAs whose expression decreased upon KSRP overexpression is listed in Supplementary Data 5, 6.

**RACE analysis.** To identify the full-length of primary-129-1, 5′ and 3′ RACE, reactions were performed on total RNA of THP-1 cells using the 5′-Full RACE Kit and 3′-Full RACE Kit (TaKaRa, Dalian, China) according to the manufacturer's manual. The primers used for the RACE experiment are listed in Supplementary Data 7.

**Luciferase reporter assay.** For functional mechanistic analyses of miR-129-1, we predicted several miR-129-1 targets using the bioinformatics software Targetscan and Miranda and validated the candidates with the luciferase reporter assay. 293TN cells were co-transfected with 0.4 μg of the reporter construct, 0.015 μg of the pRL-TK control vector and 5 pmol of the miR-129 mimic or negative control mimic. After 48 h, the cells were harvested and assayed with the Dual Luciferase Assay Kit (Promega, Madison, WI), according to the manufacturer's instructions. All transfection assays were performed in triplicate.

**In vitro processing assay.** The processing assay was performed using a 10 mg/mL HeLa cell NE. For the depletion of KSRP, ten microliters of KSRP antibody were added to 1 mg of HeLa cell NE and an IgG antibody was added as the control. After a 30-min incubation at 4 °C, the KSRP protein and antibody complex was immunoprecipitated with Protein A beads (Thermo Fisher Scientific, San Jose, CA). The supernatant is the KSRP-depleted nuclear extract (ΔKSRP-NE). The 700-nt, isotope-labeled wild-type pri-129-1 (129_WT) and pri-129-1 containing all four KSRP-binding site mutants (129_Mut) were prepared by standard in vitro transcription with T7 RNA polymerase in the presence of [α-$^{32}$P]-UTP using the 129_WT and 129_Mut constructs. 129_WT and 129_Mut were incubated with HeLa cell NEs (OriGene Technologies, Inc., Rockville, MD) for 30 min at 37 °C in the presence of KSRP (0, 1 (1×), 2 (2×), and 4 pmol (4×)). The reaction mixtures were subjected to phenol:chloroform extraction, precipitation and denaturing gel electrophoresis, followed by autoradiography.

The rescue processing assay was performed. Briefly, 20 μL of ΔKSRP-NE was incubated with 1 pmol of [α-32P]-UTP-labeled primary miR-129 and different amounts of KSRP (0.5 μg) for different times (15, 40, and 60 min) at 37 °C. The reaction mixtures were treated with proteinase K for 30 min at 37 °C, and then subjected to phenol:chloroform extraction, precipitation and denaturing gel electrophoresis, followed by autoradiography. All uncropped RNA gels can be found in Supplementary Fig. 12.

**Northern blot.** A total of 40-μg RNA samples were loaded and run on denaturing (urea) 12% polyacrylamide gels and then transferred onto Hybond membranes (Amersham Pharmacia Biotech, Piscataway, NJ). Hybridization was performed with digoxigenin (DIG)-labeled pri-miR-129, γ-$^{32}$P-labeled pre- or mature miR-129-5p probe overnight at 42 °C. U6 snRNA and rRNA were served as the loading controls for pre-, mature or primary miRNA, respectively. Expression level was measured by normalization to the expression of U6 snRNA or rRNA. The oligonucleotide probe sequences are listed in Supplementary Data 7. Expression of

miRNA was quantified by Image J software. All uncropped Northern blot can be found in Supplementary Fig. 12.

**RNA immunoprecipitation assay.** Equal amounts of THP-1 or NB4 cells were harvested in ice-cold PBS. Next, the cell pellets were resuspended in 1 mL of lysis buffer B (50 mM Tris, pH 7.4, 150 mM NaCl, 0.05% Igepal, 1 mM PMSF, 1 mM aprotinin, 1 mM leupeptin and 2 mM vanadyl ribonucleoside complexes (VRC)) and gentle sonication. After the lysates were centrifuged at 12,000 rpm for 15 min, the supernatants were precleared with Dynabeads (Invitrogen, Carlsbad, CA) in lysis buffer B with 10 μg of yeast tRNA (Sigma-Aldrich, Deisenhofen, Germany). Then, the precleared lysates were used for RIP with either an anti-KSRP antibody (Epitomics, Abcam, Cambridge, MA; 6994-1; 1:500) or a rabbit isotype IgG (12-370, Merck Millipore, Germany; 1:500) as the control for 4 h at 4 °C. The beads were washed three times with lysis buffer B, with the last wash containing an additional 0.5% sodium deoxycholate, followed by extraction with buffer C (100 mM Tris, pH 6.8, 4% SDS, 12% β-mercaptoethanol and 20% glycerol) at room temperature for 10 min. The immunoprecipitated RNA was extracted with Trizol and phenol-chloroform. For RT-PCR, each RNA sample was treated with DNase I (Ambion, DNA-free TM kit). Then, equal amounts were reverse transcribed with the High Capacity cDNA Reverse Transcription Kit (Invitrogen, Carlsbad, CA). The fold enrichment of the RNAs by KSRP was measured by qPCR and agarose gel electrophoresis. The primers used for the RIP-PCR experiments are listed in Supplementary Data 7.

**Immunoprecipitation of the protein-RNA complexes.** Maltose-binding protein (MBP)-affinity purification was used to identify proteins associated with primary miRNAs. The MS2-MBP expression plasmid was a gift from Dr. Lingling Chen (Chinese Academy of Sciences) and MS2-MBP was expressed and purified from *E. coli* using a protocol from the Steize lab (Yale University). Three MS2 coat protein-binding sites (5′-cgtacaccatcagggtacgagctagcccatggcgtacaccatcagggtacgactagta-gatctcgtacaccatcagggtacg-3′) were inserted downstream of the pri-129-1 gene sequence or the pri-129-1 KSRP-binding site mutant sequences. 293TN cells were transfected with MS2-containing constructs to obtain proteins associated with primary miRNAs, and 10 million cells were used for each immunoprecipitation assay. The cells were harvested 48 h post-transfection and subjected to RNA pull-down analysis. In brief, cells were rinsed twice with ice-cold PBS before harvesting in 10 mL ice-cold PBS by scraping. Then the cell pellets were resuspended in 1 mL IP buffer (50 mM Tris, pH 7.4, 150 mM NaCl, 0.05% Igepal, 1 mM PMSF, pro-teinase inhibitor cocktail), and subjected to 2 rounds of gentle sonication and centrifuged to obtain cell extracts. The supernatant was first incubated with MS2-MBP for 1 hr and then amylose resin beads (NEB) for another 1 hr at 4 C. The beads were washed five times with IP buffer with gentle rock and a last wash with IP buffer supplied with additional 150 mM NaCl. The protein components were finally eluted from beads with IP buffer supplemented with 12 mM Maltose, and were checked by WB with the following primary antibodies: rabbit anti-Drosha (ab12286, Abcam; 1:1000), rabbit anti-DGCR8 (ab191875, Abcam; 1:1000), rabbit anti-KSRP (6994-1, Epitomics; 1:1000,) and mouse anti-GAPDH (60004-1-Ig, Proteintech; 1:1,000,000).

**RNase H protection assay.** A standard RNase H protection reaction was performed in a total volume of 25 μL containing 1 pmol of KSRP protein, 1 pmol of [α-$^{32}$P]-UTP-labeled RNA, 20 μg/mL DNA oligonucleotides, 12 mM HEPES (pH 8.0), 60 mM MgCl₂, 1 mM DTT, 20 U of RNasin (40 U/μL, Promega, Madison, WI), and 1 U of *E. coli* RNase H. For the control, a reaction with the corresponding amount of deproteinized [α-$^{32}$P]-UTP-labeled RNA and the same ionic conditions as the extract was prepared and incubated at 25 °C for 15 min. The reaction was terminated by adding 75 μL of distilled H₂O, 100 μL of 2 × PK buffer, and 4 μL of proteinase K (10 mg/mL), and RNA was prepared as described in the 'qPCR' to analyze the RNA fragments. RNA was resuspended in 25 μL of urea gel-loading buffer (5 mM EDTA, 10% (v/v) glycerol, 0.05% (w/v) Xylene Cyanol FF, 0.05% (w/v) Bromophenol Blue, and 50% deionized formamide) and separated on urea denaturing gels, followed by autoradiography.

**qPCR.** Total RNA was extracted from the cells and tissues using Trizol reagent (Invitrogen, Carlsbad, CA) followed by DNase I digestion, according to the manufacturer's instructions. The RNA was quantified by measuring the absorbance at 260 nm and reverse transcribed. Quantitative real-time PCR (qPCR) was performed using the Bio-Rad IQ5 real-time PCR system according to the manufacturer's instructions. Primers used for reverse transcription and qPCR were shown in Supplementary Data 7. The data were normalized using endogenous *GAPDH* mRNA and U6 snRNA. The $2^{-\Delta\Delta CT}$ method was used to analyze the PCR data.

**Flow cytometry.** The cells were harvested at the indicated time points and washed twice with PBS/0.5% BSA at 4 °C to block the Fc receptors. The transduced CD34[+] HPCs were stained with APC-conjugated anti-CD14 (clone: 61D3) or anti-CD11b (clone: ICRF44) antibodies (eBioscience, San Diego, CA). Mouse BM cells were stained with a PE-conjugated anti-CD33 antibody (clone: HIM3-4 eBioscience, CA, USA). Flow cytometry was performed on a C6 Flow Cytometer (BD Biosciences,

Franklin Lakes, NJ). The data were analyzed using FlowJo software (Version 7.6, TreeStar, Ashland, OR).

**Mice and transplantation assays**. All animal experiments were performed with the approval of the Research Ethics Committee of Peking Union Medical College. Ten million CD34[+] HPCs infected with lenti-sh-KSRP, lenti-129 or lenti-GFP control were washed and resuspended in PBS and then injected into the lateral tail vein of 4-week to 6-week-old female NOD/SCID recipient mice irradiated with x-rays at a dose of 250 cGy. Four weeks after the transplantation of the CD34[+] HPCs, the mice were sacrificed, and the BM and spleens were collected and processed into single-cell suspensions for the subsequent experiments. The results were obtained from 4 mice per group.

**Western blot**. Whole-cell lysates were subjected to Western blot analysis. Rabbit polyclonal anti-KSRP antibody (6994-1; 1:1000) and anti-RUNX1 antibody (ab35962; 1:1000) that from Epitomics (Abcam, Cambridge, MA), or a GAPDH antibody (60004-1-Ig, Proteintech, Rosemont, IL; 1:100,000) were used as the primary antibodies and incubated in 1× TBST containing 5% BSA for 2 h at room temperature (RT) with continuous shaking. After washes with 1× TBST for 15 min at RT, the membranes were incubated with an HRP-conjugated goat anti-rabbit secondary Ab (Thermo Fisher Scientific, San Jose, CA) for 1 h at RT. After a final washing step with 1× TBST for 15 min at RT, the immunoreactive bands were visualized by enhanced chemiluminescence (ECL) (Amersham Biosciences, Piscataway, NJ) at indicated exposure time ("Exposure time for Western blot"). All uncropped western blots can be found in Supplementary Fig. 11.

**In vivo processing assay in zebrafish**. Embryos were collected from a laboratory breeding colony of Tuebingen zebrafish housed at 28 °C on a 14:10 h light/dark cycle and raised under standard conditions (China Zebrafish Resource Center). *KSRP* mRNA and GFP-Pri-129 WT or KSRP-binding site mutant mRNAs were synthesized in vitro from corresponding linearized plasmids using the mMESSAGE mMACHINE Kit (Thermo Fisher Scientific, San Jose, CA). The *KSRP* mRNA and GFP-Pri-129_WT mRNA (or GFP-Pri-129_Mut mRNA) were co-injected into one-cell stage zebrafish embryos. The embryos were staged at 28 °C according to the h.p.f. and morphological criteria[52]. Images of embryos staged at 4 h.p.f. and 24 h.p.f. were acquired using a Zeiss SteReo Lumar V12 (Carl Zeiss Microscopy GmbH, Jena, Germany).

**In vivo processing assay in 293TN cells**. 293TN cells were cultured in 6-well plates for 24 h until they reached ≈70–80% confluence. The cells were co-transfected with the EGFP fusion Pri-129_WT or mutant Pri-129 plasmids, pcDNA4-RFP and the pCMV-KSRP plasmid using Lipofectamine 2000 (Invitrogen, Carlsbad, CA). Twenty-four hours later, the cells were analyzed under a Zeiss Fluorescence Microscope at 400× magnification (Carl Zeiss Microscopy GmbH). RNA was also extracted for a qPCR analysis of processing efficiency by detecting the primary miR-129 vs. the mature miR-129. The fluorescence intensity was quantified using Image-Pro Plus 6.0 software (Media Cybernetics, Rockville, MD).

**In situ proximity ligation assay**. In situ PLAs were done following manufacture's protocol (Sigma-Aldrich). Rabbit anti-Drosha (ab12286, Abcam; 1:50) and rabbit anti-DGCR8 (ab191875, Abcam; 1:50) was paired with anti-KSRP (MBS246006, Mybiosource, Vancouver, Canada; 1:50), or normal rabbit IgG polyclonal antibody (12–370, Merck Millipore, Germany; 1:50) for the primary antibodies. Anti-KSRP and normal rabbit IgG polyclonal antibody as a negative control. Before blocking, 293T or HeLa cells were grown on chambered slides (Sigma), fixed in 10% formaldehyde, permeabilized with 0.25% Triton X-100-PBS. Cells were analyzed by Zeiss LSM780 confocal immunofluorescence microscopy (Zeiss, Germany) followed by deconvolution with Huygens Essential version 16.05 (Scientific Volume Imaging, The Netherlands, http://svi.nl). PLA dots per cell were quantified with BlobFinder V3.2 image analysis software (Uppsala University, Stockholm, Sweden) and the mean number of PLA dots per cell was calculated by analyzing a minimum of 100 cells.

**Bioinformatics analysis**. All paired-end reads were trimmed for adaptor sequence using Trim Galore (Version 0.3.7) with parameters "-q 25 --stringency 5 --length 50 --paired --phred33" and filtered for low-quality reads. The hg38 reference genome (GRCh38) and gene annotation file (GTF format) was downloaded from GENCODE Release 24 (http://www.gencodegenes.org/)[53]. TopHat[54,55] (Version 2.1.0) was used to align the filtered reads to the hg38 genome reference genome with default settings, and FPKMs (Fragments Per Kilobase transcriptome per million reads) were obtained using Cufflinks[54] (Version 2.2.1) with default parameters. Expression levels were estimated using feature Counts[56] (Version 1.5.0) with parameters "-T 7 --primary", gene-level raw counts were derived from the sum of the corresponding values for all isoforms of a gene.

EBSeq-HMM[18] were used to analyze the differences in gene expression among three stages of monocyte and granulocyte differentiation, respectively. There are two main steps for EBSeq-HMM to get differentially expressed (DE) genes:

1) Detecting *DE* genes that showed significant change between any two stages under a target false discovery rate (FDR) of 0.05.
2) Clustering *DE* genes into expression paths with the maximum posterior probability (PP) greater than 0.5.

By using EBSeq-HMM, we obtained DE genes and its corresponding expression paths (i.e., "Up-Down") at day 5, 10 and 15 of monocyte (or granulocyte) differentiation (Supplementary Data 1). The expression path "Up-Down" represents that a gene was up-regulated at day 10 compared with day 5 while downregulated at day 15 compared with day 10. Monotonically increased ("Up-Up") or decreased ("Down-Down") genes, during monocyte (or granulocyte) differentiation were chosen for further analysis.

Gene expression levels were estimated using the FPKM, and all of the genes with the FPKM ≥ 1 as the expressed genes were used for further analysis. Information of RBP was downloaded from RBPDB19 and ATtRACT20 databases. We developed an R script (Supplementary Data 8) to classified *DE* genes with "Up-Up" or "Down-Down" expression paths (Supplementary Data 2, 3) into different expression patterns.

The heat maps were drawn by using the function 'pheatmap' of R packages 'pheatmap' and the VennDiagram were drawn by using the function 'draw.pairwise.venn' of R package 'VennDiagram'. K-means clustering was performed with the output of function 'pheatmap' and clusters were obtained with function 'cutree' by using R.

**Plasmid electrotransfections**. We used the Neon Transfection System according to the manufacturer's instructions (Invitrogen, Carlsbad, CA) to improve the plasmid transfection efficiency in THP-1 cells for the RNA-seq analysis. Cells were electrotransfected with plasmids (pEGFP, pSIH-H1-sh-luc, pEGFP-KSRP or pSIH-H1-sh-KSRP). The electroporation parameters for $1 × 10^7$ THP-1 or NB4 cells were a 1350-V pulse, 10-ms pulse width, and 4 pulses using a 10-μL tip. Forty-eight hours later, the electrotransfection efficiency was detected by Western blotting or qPCR.

**Rescue assays**. To validate that KSRP promotes pri-miR-129 processing to regulate myeloid lineage differentiation, THP-1 cells were transfected with pEGFP-KSRP using Lipofectamine 2000, and 24 h later, the cells were transfected with a miR-129 inhibitor. NB4 cells were transfected with si-KSRP at a final concentration of 100 nM using DharmFECT1, and 24 h later the cells were transfected with a miR-129 mimic. To validate that miR-129 target RUNX1 to regulate myeloid lineage differentiation, THP-1 and NB4 cells were transfected with si-RUNX1 and a miR-129 inhibitor or their controls by the Neon transfection system. Cells were then stimulated with PMA or ATRA for an additional 48 h before being harvested for subsequent analyses. CD14 expression in THP-1 cells undergoing monocytic differentiation and CD11b expression in NB4 cells undergoing granulocytic differentiation were analyzed by flow cytometry.

**Exposure time for Western blot**. Most of the Western blot was developed by Tanon 5200 automatic digital gel image analysis system (Tanon, Shanghai, China), and the exposure time of GAPDH is 150-350 ms. In Fig. 2a, exposure time of samples along monocytic differentiation is 20 s for KSRP; exposure time of samples along granulocytic differentiation is 200 ms for KSRP. In Fig. 2b, exposure time is 20 s for KSRP. In Fig. 4f–h, exposure time is 60 s for KSRP. In Fig. 5b, exposure time is 60 s for Drosha, DGCR8 and KSRP. In Fig. 6j, k, exposure time is 90 s for RUNX1. In Fig. 7g, exposure time is 90 s for RUNX1 and KSRP. In Fig. 7h, exposure time is 30 s for RUNX1 and KSRP. In Supplementary Fig. 2a, exposure time of samples along monocytic differentiation is 3 s for KSRP; exposure time of samples along granulocytic differentiation is 300 ms for KSRP. In Supplementary Fig. 4a, e, f, exposure time is 60 s for KSRP. In Supplementary Fig. 7m, exposure time is 10 s and 90 s for RUNX1 in THP-1 and NB4, respevtively. In Supplementary Fig. 8a, exposure time is 60 s for RUNX1.

For other pictures showing the result of manual exposure (Fig. 2d, i; Supplementary Fig. 7n), exposure time for GAPDH is estimated 1 s. In Fig. 2d, exposure time is 30 s for KSRP in PMA-induced samples and 10 s for KSRP in ATRA-induced samples. In Fig. 2i, exposure time is 60 s for KSRP. In Supplementary Fig. 7n, exposure time is 90 s for RUNX1.

**Isolation of monocytes and neutrophils from peripheral blood**. Informed consent was obtained in accordance with the Institutional Review Board of the Chinese Academy of Medical Science. Human monocytes and neutrophils were isolated from the peripheral blood of healthy donors in 3% dextran and were isolated by percoll gradient centrifugation followed by purification by CD14 MicroBeads (130-050-201, Miltenyi Biotec, Bergisch-Glad-bach, Germany) or EasySep Human Neutrophil Enrichment Kit (#19257, Stemcell Technologies, Canada). After sorted by CD14 microbeads, the obtained CD14 positive cells were plated into T-25 flasks and cultured for 4-6 h at 37 °C until the adherent monocytes were collected by scraping.

**Statistical analysis**. For all studies, *n* per group is as indicated in the figure legend. All data were analyzed using GraphPad Prism 5.0 software (GraphPad Software,

Inc., USA) and presented as means ± SD. Differences between experimental groups and control groups were analyzed using the two-tailed Student's t-test. Statistical significance was accepted at $P < 0.05$. No statistical methods were used to pre-determine sample size. All experiments were carried out with at least three biological replicates. We chose the appropriate tests according to the data distributions. The experiments were not randomized and we did not exclude any samples. The investigators were not blinded to allocation during experiments and outcome assessment.

**Data availability**. miRNA and pri-miR-RNA sequencing data have been deposited in the GEO database under accession code GSE87318. The data for mRNA sequencing have also been deposited in the GEO database under accession code GSE87088. Source data for Fig. 1c–e have been provided as Supplementary Data 1–4. Source data for Fig. 2b, c have been provided as Supplementary Data 5, 6. All other data supporting the findings of this study are available from the corresponding author on request.

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

## Acknowledgements

This work was supported by the National Key Research and Development Program of China (2016YFA0100601 to J.Y., 2016YFA0100103 to Y.M.), the National Key Basic Research Program of China (2015CB943001 to J.Y.), the National Natural Science Foundation of China (31371322, 81530007 and 91440111 to J.Y.; 31471227 to F.W.; 31571523 to Y.M.; 81770104 to D.W.), and CAMS Innovation Fund for Medical Sciences (2017-I2M-3-009 to X.W. and J.Y.).

## Author contributions

H.Z. and X.W. designed and performed the experiments, P.Y., Y.S., J.H., S.Y., Y.R. and Y. M. performed the experiments. P.T. and D.W. performed the RNA-seq analysis. J.Z. supervised cell culture experiments and reviewed the manuscript. J.Y. and F.W. planned the research strategy and designed the experiments. J.Y. and X.W. wrote the paper.

## Additional information

**Competing interests:** The authors declare no competing financial interests.

