## [Peer Review File · Nature Communications]

Reviewers' Comments:

Reviewer #1 (Remarks to the Author)

In this well-performed study, the researchers pick an RNA-binding protein (RBP) named KSRP from their time course analysis of CD34+ HPC-derived monocyte and granulocyte differentiation given their opposite level changes in these two differentiation processes. They are able to demonstrate that KSRP promotes the processing of miR-129, which in turn blocks the expression of the RUNX1. As aforementioned, the study is well performed, control experiments are correct and the axis KSRP-miR-129-RUNX1 is well drawn. Just a few comments:

- It would be important to determine the levels of both the KSRP protein, and the miR-129 in mature peripheral blood human monocytes and granulocytes to define the reach of the author's findings regarding the regulatory role of KSRP axis in monocytic and granulocytic differentiation *in vivo*.

- Differentiation experiments with primary cells are done with human umbilical cord blood. Did the author get the same results regarding the levels of KSRP and miR-129 using CD34+ cells from bone marrow, the normal source of monocytes and granulocytes?

- The authors mentioned a second miRNA (specifically miR-140) with inverse expression patterns for their pri-miRs in NB4 cells undergoing differentiation. Why did they discard miR-140? Any interesting targets for these miRNA?

Reviewer #2 (Remarks to the Author)

In this manuscript, the authors comprehensively demonstrated that KSRP plays distinct roles in regulating monocytopoiesis and granulopoiesis through directly impacting miR-129 biogenesis. Moreover, they showed that miR-129 controls the fate of myeloid differentiation through targeting Runx1, a member of CBF family of proteins that was previously shown to transactivate M-CSFR expression while inhibiting G-CSFR expression. These data are novel and quite interesting. Overall, I view the manuscript favorably, given a thorough mechanistic study performed. I do however have some concerns that would need to be addressed prior to giving my full endorsement.

Specific experimental points:

1. In Fig.2, the authors demonstrated that KSRP was downregulated during monocytic differentiation but upregulated during granulocytic differentiation. If this is the case, it is unclear as to why at day 3 after M-CSF or G-CSF treatment, the level of KSRP in CD34+ HPCs was so much higher under monocytic differentiation condition than that in cells under granulocytic differentiation condition (Fig.2a). How did the levels of KSRP alter in those CD34+ HPCs during two different myeloid cell differentiations between day 3 and 0 hr time point (e.g. 0 hr, 24 hr, 48 hr and 72 hr similar to what were shown in Fig.2b)?

2. As the authors also acknowledged that KSRP has been previously shown to positively promote a specific subset of miRNAs (Trabucchi et al. 2009). However, there is very little overlap between miRNAs that are listed in this manuscript vs. the ones shown in the previous report. A proper discussion should be included if it cannot be addressed experimentally.

3. In Fig.6b, there were barely any GFP+ cells that could be detected in cells treated with miRZIP-129. I am not sure whether it is fair to compare those so-called GFP+ cells to other GFP+ cells. The amounts of miR-129 in GFP+ cells with or without miRZIP-129 (or lenti-129) need to be shown.

4. Likewise, it seems that transfection of KSRP and/or miR-129 inhibitor greatly influenced the frequencies of GFP+ cells (Fig.S6i). The authors should examine myeloid differentiation from the GFP- populations in all experimental groups. Since KSRP (or miR-129) levels should not be altered in those GFP- cells, one would expect that there should be no difference in terms of CD14+ in monocytic differentiation condition (or CD11b+ cells in granulocytic differentiation condition) between different groups.

5. To directly examine the role miR-129-mediated Runx1 repression during myeloid differentiation, the authors need to perform rescue experiments similar to what they did in Fig.S6i-l. To this end, cells should be co-transfected/transduced with miR-129 plus lenti-Runx1 or miR-129 inhibitor plus si-Runx1 during monocytic and granulocytic differentiation, respectively.

Reviewer #3 (Remarks to the Author)

To study myeloid cell specification, authors differentiated human CD34+ hematopoietic progenitors from umbilical cord blood and differentiated them into monocytes or granulocytes by adding M-CSF or G-CSF, respectively. They performed RNA-seq at three time points (days 5, 10 and 15) for each culture and focussed on differentially expressed RNA-binding proteins since the role of post-transcriptional regulation is poorly understood. They identified and validated KSRP to be preferentially up-regulated late during granulocytic differentiation. They found that KSRP can bind the primary-miR-129-1 transcript via three binding sites, and serves to enhance production of mature miR-129-1. In turn, miR-129-1 represses RUNX1 to promote granulocyte differentiation at the expense of monocytes.

There are at least two major points where I feel the authors have over-interpreted their data:

1) Since KSRP is up-regulated late, it is not easy to imagine how it controls the granulocyte vs monocyte cell fate decision which should occur earlier. KSRP seems to be a good marker of granulocyte differentiation, but its timing of expression is not consistent with a role in lineage specification.

2) There is no direct evidence that KSRP interacts in vivo with Drosha in the nucleus and Dicer in the cytoplasm. The authors could try a proximity ligation assay to support these claims.

General comments:

KSRP was previously reported to inhibit proliferation. Did authors look at cellular proliferation in their myeloid differentiation experiments?

The authors propose a simplistic pathway (KSRP-miR-129-RUNX1); however, they should acknowledge/discuss more that in reality KSRP's role may be more complicated since it can bind other microRNAs, and mRNAs as well. KSRP has been reported to play roles in RNA splicing and mRNA decay.

Specific comments on figures:

Fig 2d, the knock-down of KSRP appears to be modest (although statistically significant). The greatest knock-down is only apparent on day 20 of granulocyte differentiation. Authors should employ another shRNA to knock-down KSRP in case this is due to an off-target effect. Authors should also use a scrambled or irrelevant shRNA as a negative control rather than empty vector.

Fig 2e-f, Fig 6a-b, and Fig 6n, please compare % differentiated monocytes/granulocytes between untransduced (gate on GFP-) vs transduced (gate on GFP+) cells. Since CD11b is also expressed on monocytes, dot plots of CD11b vs CD14 should also be shown.

Fig 3d, One cannot tell whether signal is from RNA and/or DNA contamination since there is no -RT control. Some samples from differentiating granulocytes should be included to gauge the level of over-expression

Fig 3g, there are no size markers for Northern blot. What is the size of band labelled as pri-miR-129? Furthermore, mature, pre- and pri-miR-129-1 should be quantified.

Fig 3, genome browser screenshot of RNA-seq coverage track should be shown for pri-miR-129-1 vs -2

Fig 3h, it is not clear from Methods how pre-miR-129-1 was assayed

Fig 3i, on which day of culture were assays performed?

Fig 4j-k, The quantitation does not seem to match the autoradiograph. Perhaps a shorter exposure would help. The degree of protection is very modest.

Fig 5h, GFP signal needs to be quantified; could also perform flow cytometry. How do authors control for differences in transfection efficiency among constructs?

Fig 6, lenti-GFP is not a good control for lenti-miR-129 or lenti-Zip-129.

minor comments:

Fig1a - differentiated cells don't look like Granulocytes which typically have lobed/segmented nuclei

The font size in some figures are frequently hard to read (too small)

a few typos

Response to the referees

Reviewers' comments:

Reviewer #1 (Remarks to the Author):

In this well-performed study, the researchers pick an RNA-binding protein (RBP) named KSRP from their time course analysis of CD34+ HPC-derived monocyte and granulocyte differentiation given their opposite level changes in these two differentiation processes. They are able to demonstrate that KSRP promotes the processing of miR-129, which in turn blocks the expression of the RUNX1. As afore mentioned, the study is well performed, control experiments are correct and the axis KSRP-miR-129-RUNX1 is well drawn. Just a few comments:

- It would be important to determine the levels of both the KSRP protein, and the miR-129 in mature peripheral blood human monocytes and granulocytes to define the reach of the author's findings regarding the regulatory role of KSRP axis in monocytic and granulocytic differentiation in vivo.

Response: As the reviewer suggested, monocytes and granulocytes were sorted from normal human peripheral blood by CD14 MicroBeads (130-050-201, Miltenyl Biotec) and EasySep Human Neutrophil Enrichment Kit (#19257, Stemcell Technologies) after standard density gradient separation, respectively. RNA was extracted and reverse transcribed. Relative KSRP and miR-129 expression was examined in these cells by qPCR. Out of 3 people enrolled, both KSRP and miR-129 were significantly higher in granulocytes than that in monocytes in 2 people, and no significant changes of KSRP and miR-129 were found between monocytes and granulocytes in the other 1 people (Figure R1). We speculate that expression of KSRP is different among individuals and subjects to the physiological condition of one person (eg. infection). Overall, our result indicates that KSRP positively regulates the levels of miR-129 in mature peripheral blood cells, and more important, higher KSRP and miR-129 tend to favor granulocyte maturation while lower KSRP and miR-129 expression tend to favor monocyte maturation.

Figure R1. RT-qPCR analysis of KSRP mRNA and miR-129 levels in monocytes and granulocytes from normal human peripheral blood.

- Differentiation experiments with primary cells are done with human umbilical cord blood. Did the author get the same results regarding the levels of KSRP and miR-129 using CD34+ cells from bone marrow, the normal source of monocytes and granulocytes?

Response: As the reviewer suggested, we investigated the expression of both KSRP and miR-129 during monocytic or granulocytic differentiation of bone marrow derived CD34⁺ cells. Western blot of KSRP showed that KSRP was increased following granulocytic induction while decreased along monocytic differentiation dramatically at day 10 (Figure. R2, left panel. also see Fig. 2b). qPCR result showed the similar expression pattern of KSRP mRNA and miR-129 (Figure. R2, middle and right panel. also see Supplementary Fig. S2a; Fig. 3i). These results indicate that KSRP and miR-129 not only regulate fetal myeloid differentiation but also affect the differentiation of adult monocytes and granulocytes, which is critical to sustain functional immune responses.

Figure 2b

Figure S2a

Figure 3j

Figure. R2. The expression of KSRP and miR-129 in bone marrow CD34⁺ differentiation.

- The authors mentioned a second miRNA (specifically miR-140) with inverse expression patterns for their pri-miRs in NB4 cells undergoing differentiation. Why did they discard miR-140? Any interesting targets for these miRNA?

Response: Although miR-140 was upregulated and pri-miR-140 was downregulated during granulocytic differentiation of NB4 cells, but we failed to confirm the regulation of pri-miR-140 by KSRP. As shown in Fig. 3e, both miR-140 and pri-miR-140 were only slightly changed after KSRP expression modulation, and miR-140 even sees an upregulated tendency when KSRP was knockdown. Due to this reason, miR-140 was not taken into consideration when selecting candidates. However, it is possible that miR-140 can also divergently regulate monocytic and granulocytic differentiation, which needs our further investigation.

Reviewer #2 (Remarks to the Author):

In this manuscript, the authors comprehensively demonstrated that KSRP plays distinct roles in regulating monocytopoiesis and granulopoiesis through directly impacting miR-129 biogenesis. Moreover, they showed that miR-129 controls the fate of myeloid differentiation through targeting Runx1, a member of CBF family of proteins that was previously shown to transactivate M-CSFR expression while inhibiting G-CSFR expression. These data are novel and quite interesting. Overall, I view the manuscript favorably, given a thorough mechanistic study performed. I do however have some concerns that would need to be addressed prior to giving my full endorsement.

Specific experimental points:

1. In Fig.2, the authors demonstrated that KSRP was downregulated during monocytic differentiation but upregulated during granulocytic differentiation. If this is the case, it is unclear as to why at day 3 after M-CSF or G-CSF treatment, the level of KSRP in CD34⁺ HPCs

was so much higher under monocytic differentiation condition than that in cells under granulocytic differentiation condition (Fig.2a). How did the levels of KSRP alter in those CD34+ HPCs during two different myeloid celldifferentiations between day 3 and 0 hr time point (e.g. 0 hr, 24 hr, 48 hr and 72 hr similar to what were shown in Fig.2b)?

Response: The Western Blot shown in Fig. 2a was performed separately on two membrane, therefore, it is hard to conclude that the level of KSRP was higher in monocytic-differentiated CD34⁺ HPCs than that in granulocytic-differentiated cells since the exposure time and experiment conditions were not the same. To further investigate whether KSRP was upregulated during granulocytic differentiation and downregulated during monocytic differentiation from the beginning of induction, cells at 0 hr, 36 hr and 72 hr were collected and expression of KSRP was examined by one Western Blot. Due to the limited number of CD34⁺ cells, we only use 3 time points between day 3 and 0 hr. As shown in below Figure. R3, expression of KSRP was almost unchanged between day 3 and 0 hr during granulocytic differentiation but down-regulated during monocytic differentiation. After then its expression was increased during granulocytic differentiation but decreased during monocytic differentiation, indicating that KSRP expression was changed after 3 day of differentiation. This new result was added to the new Supplementary Fig. S2a.

Figure R3. Immunoblot of KSRP expression at 0 hr, 36 hr and 72 hr of CD34⁺ HPC granulocytic and monocytic differentiation.

2. As the authors also acknowledged that KSRP has been previously shown to positively promote a specific subset of miRNAs (Trabucchi et al. 2009). However, there is very little overlap between miRNAs that are listed in this manuscript vs. the ones shown in the previous report. A proper discussion should be included if it cannot be addressed experimentally.

Response: Indeed, Trabucchi's work showed that KSRP knockdown decreased 14 miRNA levels >1.5 fold as determined by microarray, and out of which 6 was confirmed by Northern blot. They also found several miRNAs whose expression were unaffected. However, by

miRNA-seq we identified 16 significantly upregulated miRNAs upon KSRP overexpression with only three overlaps (let-7d, miR-98 and miR-16) with results of Trabucchi et al.. It could be noticed that Trabucchi and we used different method to detect KSRP-regulated miRNA candidates (microarray vs. miRNA seq). It is also known that compared with microarray, RNA-seq is relatively more sensitive since it delivers low background signal. With array hybridization technology, gene expression measurement is limited by background at the low end and signal saturation at the high end. Nevertheless, RNA-Seq quantifies discrete, digital sequencing read counts, offering a broader dynamic range. More importantly, we used THP-1 cells instead of HeLa cells for screen. Until now many miRNAs have been characterized as tissue-specific in mammals, and that's why these class of RNAs are important for development and disease regulation. It could be speculated that most miRNAs we found are particularly regulated by KSRP in myeloid cells. Furthermore, another issue arises from the difference in cell type is that diverse cell-specific co-factors of KSRP may exist, participating in miRNA processing. To be more specific, in another paper (FASEB J. 2009 Sep;23(9):2898-908.) from the same lab showed that maturation of miR-155, a miRNA that was not included in the mentioned paper (Trabucchi et al. 2009), could be favored by KSRP. However, this work was done in bone marrow-derived macrophages (BMDMs) under inflammatory condition. Overall, although mechanically KSRP promotes the processing of miRNAs, the variety of miRNAs it affects depends on cell types, physiological conditions and are still largely unknown due to technical limitations. The above discussion has been added into the revised manuscript.

3. In Fig.6b, there were barely any GFP+ cells that could be detected in cells treated with miRZIP-129. I am not sure whether it is fair to compare those so-called GFP+ cells to other GFP+ cells. The amounts of miR-129 in GFP+ cells with or without miRZIP-129 (or lenti-129) need to be shown.

Response: We first apologized for the confused presentation of results in Figure 6b. By adjusting experiment conditions, the percentage of GFP+ cells in miRZip-129 transduced cells was increased to 19.9%. To confirm that miR-129 expression was indeed knocked down in these GFP+ cells, the GFP+ and GFP- cells were sorted after miRZip-129 or control transduction, respectively. miR-129 expression in GFP+ and GFP- cells was analyzed by qPCR. As shown in the below Figure. R4, the decrease in miR-129 level was exclusively observed in GFP+ cell populations upon miRZip-129 transduction. This results was added to the new Supplementary Fig.S6a, b.

Figure. S6a

Figure S6b

Figure. R4. CD34⁺ HPCs were transduced with lenti-129 or lenti-control, or miRZIP-129 or miRZIP-control for 24 h, and then cultured for 15 days to allow cells to differentiate into monocytes (S6a) or granulocytes (S6b). Relative miR-129 expression in GFP⁺ and GFP⁻ population from miRZIP-129 or miRZIP-control transduced HPCs were analyzed by qPCR on day 15.

4. Likewise, it seems that transfection of KSRP and/or miR-129 inhibitor greatly influenced the frequencies of GFP⁺ cells (Fig.S6i). The authors should examine myeloid differentiation from the GFP⁻ populations in all experimental groups. Since KSRP (or miR-129) levels should not be altered in those GFP⁻ cells, one would expect that there should be no difference in terms of CD14⁺ in monocytic differentiation condition (or CD11b⁺ cells in granulocytic differentiation condition) between different groups.

Response: As the reviewer suggested, this rescue experiment was repeated in THP-1 cells. Percentages of CD14⁺ cells were analyzed in both GFP⁺ and GFP⁻ population. As expected, % CD14⁺ cells was almost unchanged in GFP⁻ population between the 4 groups. However, % CD14⁺ cells were decreased by KSRP overexpression and restored after co-transfection of miR-129 inhibitor (Figure R5.). These results are shown in the new Supplementary Fig. S7i.

Figure S7i

Figure S7j

Figure R5. Rescue assay in THP-1 cells.

5. To directly examine the role miR-129-mediated Runx1 repression during myeloid differentiation, the authors need to perform rescue experiments similar to what they did in Fig.S6i-l. To this end, cells should be co-transfected/transduced with miR-129 plus lenti-Runx1 or miR-129 inhibitor plus si-Runx1 during monocytic and granulocytic differentiation, respectively.

Response: As the reviewer suggested, we performed rescue experiment by co-transfecting miR-129 inhibitor plus si-Runx1 in THP-1 and NB4 cells followed by monocytic or granulocytic differentiation. As shown in below Figure R5 (also see Fig. S8a), si_RUNX1 restored the increase of endogenous RUNX1 protein induced by miR-129 inhibitor (anti-129). After PMA or ATRA induction, cotransfection significantly rescued miR-129 inhibitor-stimulated CD14 increase as well as CD11b/CSF3R reduction (Figure R6. also see Fig. S8b and S8c). These results were added to the new Supplementary Figure. S8.

Figure. S8a

Figure. S8b

Figure. S8c

Figure. R6. The rescue assay was performed by transfecting THP-1 or NB4 cells with a combination of miR-129 inhibitor (or control) plus si-Runx1 (or control), followed by monocytic or granulocytic induction for 48 h.

Reviewer #3 (Remarks to the Author):

To study myeloid cell specification, authors differentiated human CD34+ hematopoietic progenitors from umbilical cord blood and differentiated them into monocytes or granulocytes by adding M-CSF or G-CSF, respectively. They performed RNA-seq at three time

points (days 5, 10 and 15) for each culture and focussed on differentially expressed RNA-binding proteins since the role of post-transcriptional regulation is poorly understood. They identified and validated KSRP to be preferentially up-regulated late during granulocytic differentiation. They found that KSRP can bind the primary-miR-129-1 transcript via three binding sites, and serves to enhance production of mature miR-129-1. In turn, miR-129-1 represses RUNX1 to promote granulocyte differentiation at the expense of monocytes.

There are at least two major points where I feel the authors have over-interpreted their data:

1) Since KSRP is up-regulated late, it is not easy to imagine how it controls the granulocyte vs monocyte cell fate decision which should occur earlier. KSRP seems to be a good marker of granulocyte differentiation, but its timing of expression is not consistent with a role in lineage specification.

We apologized for the improper description of our results. We have revised these words in Introduction and Discussion sections to accurately describe the function of KSRP in granulocyte and monocyte differentiation.

2) There is no direct evidence that KSRP interacts in vivo with Drosha in the nucleus and Dicer in the cytoplasm. The authors could try a proximity ligation assay to support these claims.

Response: In 2009, Trabucchi et al. demonstrated that KSRP forms complexes with Drosha or Dicer to regulate the biogenesis of a subset of miRNAs. In this paper, we also found that KSRP could be coimmunoprecipitated with Flag-tagged Drosha and DGCR8 in an RNase independent manner. These results both indicate that KSRP could directly binds to Drosha and DGCR8. In this work, we didn't focus on the miRNA processing step mediated by Dicer and the content related to Dicer has been removed. As the reviewer suggested, to further investigate the close association of KSRP with Drosha, and KSRP with DGCR8 in vivo, 293T and HeLa cells were fixed, permeabilized, reacted with anti-KSRP and anti-Flag after co-transfection of KSRP and Flag-tagged Drosha or DGCR8 into these cells. Cells that were reacted with both anti-Flag and anti-IgG were used as negative controls. As shown in below Figure. R7, significant number of proximity signals (dots) per nucleus was detected in samples incubated with anti-KSRP and anti-Flag (also see Fig. 5d; Supplementary Fig. S5c), but not in the negative ones. Additionally, proximity signals were more intense in cells co-transfected with Drosha-Flag and KSRP than that transfected with DGCR8-Flag and KSRP. Results of PLA experiment was quantified using BlobFinder V3.2 image analysis software.

Figure 5d

Figure S5c

Figure. R7. proximity ligation assay.

General comments:

KSRP was previously reported to inhibit proliferation. Did authors look at cellular proliferation in their myeloid differentiation experiments?

Response: To investigate the function of KSRP in cellular proliferation, CCK8 assay was performed either in undifferentiated cells or myeloid differentiation-induced cells. In general, NB4 or THP-1 cells were transfected with si-KSRP or the scramble control (NC). Sixteen hours later cells were treated with ATRA, PMA, or not, and proliferation was analyzed every 24 hrs. As shown below, KSRP do not affect cell proliferation in NB4 and THP-1 cells, no matter whether differentiation was induced. This result suggests that KSRP regulates myeloid cell differentiation with little or no effect on cell proliferation. However, Trabucchi et al. (Nature. 2009 Jun 18;459(7249):1010-4.), Pruksakorn D et al. (Int J Oncol. 2016 Sep;49(3):903-12.) and Malz M (Hepatology. 2009 Oct; 50(4):1130-9.) have reported that KSRP promotes cell

proliferation in osteosarcoma cell lines and hepatocellular carcinoma cells, suggesting that KSRP could regulate cell proliferation through co-activating the cell-specific class of KSRP-dependent miRNAs in these cells.

Figure. R8. CCK8 assay of NB-4 and THP-1 cells upon KSRP knockdown.

The authors propose a simplistic pathway (KSRP-miR-129-RUNX1); however, they should acknowledge/discuss more that in reality KSRP's role may be more complicated since it can bind other microRNAs, and mRNAs as well. KSRP has been reported to play roles in RNA splicing and mRNA decay.

Response: Indeed KSRP is a multifunctional post-transcriptional regulator that may have a relevant role in virtually all steps of mRNA metabolism. As reviewed by Gherzi R et al. (2014), KSRP was originally described as a nuclear factor regulating transcription, the c-src pre-mRNA splicing, or the apolipoprotein B editing. Recently, KSRP was comprehensively studied for its ability to promote mRNA decay and the biogenesis of distinct sets of miRNAs. Other functions of KSRP, including pre-mRNA splicing, translational inhibition, and mRNA export and localization were also been reported. Therefore, the KSRP-miR-129-RUNX1 pathway tend to be one of the multifold complex mechanisms involving KSRP's regulation in myeloid differentiation. We have discussed these information and accordingly added to the revised manuscript in the Discussion section.

Specific comments on figures:

Fig 2d, the knock-down of KSRP appears to be modest (although statistically significant). The greatest knock-down is only apparent on day 20 of granulocytedifferentiation. Authors should employ another shRNA to knock-down KSRP in case this is due to an off-target effect. Authors should also use a scrambled or irrelevant shRNA as a negative control rather than empty vector.

Response: To obtain better knock-down results, we purchased a new set of KSRP shRNA from OriGene (TL311984; KHSRP - Human, 4 unique 29mer shRNA constructs in lentiviral GFP vector). The non-effective 29-mer scrambled shRNA cassette in the same lentiviral vector was also provided by OriGene. The 4 shRNA was mixed to transduce cells. We found that the knock-down efficiency was significantly elevated (Figure R9, also see Fig. 2e). To further analyze the effect of KSRP knock-down on monocytic and granulocytic differentiation, differentiation markers was also examined by qPCR and flow cytometry (Figure R9, also see Fig. 2e-g). Similar to our former results, KSRP knock-down increased CD14 expression while suppressed CD11b expression from day 5 of monocytic or granulocytic differentiation. These results thus verified our proposal that KSRP divergently regulate monocytic and granulocytic differentiation. These new results were added to the new Figure 2e-g.

Figure R9. Functional analysis of KSRP knockdown in CD34⁺ HPCs.

Fig 2e-f, Fig 6a-b, and Fig 6n, please compare % differentiated monocytes/granulocytes between untransduced (gate on GFP-) vs transduced (gate on GFP+) cells. Since CD11b is also expressed on monocytes, dot plots of CD11b vs CD14 should also be shown.

Response: As the reviewer suggested, these experiments have been repeated. In granulocytic differentiation, CD11b expression in both GFP⁺ and GFP⁻ cells was analyzed. In monocytic differentiation, CD14 expression as well as percentages of CD11b⁺CD14⁺ cells were analyzed. As shown below, comparison of % differentiated monocytes/granulocytes between untransduced (gate on GFP-) and transduced (gate on GFP+) cells were shown (also

see Fig. 2f, 2g; Supplementary Fig. S6). The ratio of cells in GFP⁺ to that in GFP⁻ was shown in the right of each figure (parentheses means the normalized ratio to the control).

We found that although the percentages of CD11b and/or CD14 positive cells in GFP⁺ cells were changed upon miR-129, KSRP or RUNX1 expression alternation, the percentages of these cells in GFP⁻ cells were almost similar when compared with control groups. Moreover, the ratio of % differentiated monocytes/granulocytes between GFP⁺ vs GFP⁻ cells was increased or decreased in the same tendency with % differentiated monocytes/granulocytes in GFP⁺ cells. These results indicate that changes upon differentiation conditions in GFP⁺ (transduced) cells could reflect the effect of miR-129, KSRP and RUNX1 expression alternation.

Figure 2f

Figure 2g

Figure S6

Figure R10. The repeated FACS results in Figure 2 and Figure S6.

Fig 3d, One cannot tell whether signal is from RNA and/or DNA contamination since there is no -RT control. Some samples from differentiating granulocytes should be included to gauge the level of over-expression

Response: We apologized for the insufficient description of qPCR methods. In fact, we have included DNaseI digestion after total RNA extraction, therefore it could be reasonable to

deduce that signal is from RNA but not DNA contamination. From our understanding of the second question, the answer is as follows: in differentiating granulocytes, KSRP was upregulated as demonstrated in both HPCs (Fig. 2a, b; Supplementary Fig. S2a) and NB4 as well as HL-60 cells (Fig. 2c, d). Accordingly, we also detected the changes in both primary and mature levels of candidate miRNAs along granulocyte differentiation (Supplementary Fig. S3a).

Fig 3g, there are no size markers for Northern blot. What is the size of band labelled as pri-miR-129? Furthermore, mature, pre- and pri-miR-129-1 should be quantified.

Response: We apologized for the missing of size marker for Northern blot. The RNA marker from the same Northern blot was added to the full size picture in Supplementary Figure. 3c (Figure. R11), and according to the marker, the size of pri-miR-129 is ~700nt (~700nt is in consistent with the RACE results). Due to space limitation, cropped picture was shown in Figure 3h and the quantified results of mature, pre- and pri-miR-129 was also shown in Figure 3h by Image J software.

Figure S3c

Figure 3h

Figure. R11. The revised Northern blot results.

Fig 3, genome browser screenshot of RNA-seq coverage track should be shown for pri-miR-129-1 vs -2

Response: As the reviewer suggested, we used coverage (Wiggle) files generated from SAMTOOLS and the annotation file were loaded onto the Integrated Genome Viewer (IGV,

developed at the Broad Institute) to display the RNA-seq coverage across the miR-129-1 locus. As shown below (also see Figure 3g), the y axis shows the number of reads mapping to the location of the miR-129-1 locus (No reads were mapped to the miR-129-2 locus, therefore the data was not shown).

Figure 3g

Figure R12. Visualization of RNA-seq coverage across the miR-129-1 locus.

Fig 3h, it is not clear from Methods how pre-miR-129-1 was assayed

Response: We are sorry for the missing of primer information about pre-miR-129. We used the pre-miR-129-1 specific RT primers and Invitrogen RT kits for Reverse transcription, and the pre-miR-129-1 relative expression was detected by specific qPCR primers. RT primers and qPCR primers are shown below, which were also inserted to the revised Supplementary Table S6.

Pre-miR-129-1 RT primer: 5'-GTCGTATCCAGTGCAGGGTCCGAGGTATTTCGACTGGATACGACAGATACTTTT-3'

Pre-miR-129-1-UP: 5'-GGATCTTTTTGCGGTCTGGGCTTG-3'

Pre-miR-129-1-Down: 5'-TCCAGTGCAGGGTCCGAGGT-3'

Fig 3i, on which day of culture were assays performed?

Response: We are sorry that we made a mistake in Fig 3i since this experiment was actually performed in THP-1 cells without PMA induction. Figure 3 has been revised and the title of 3i has been changed to "THP-1".

Fig 4j-k, The quantitation does not seem to match the autoradiograph. Perhaps a shorter exposure would help. The degree of protection is very modest.

Response: The picture in Fig 4j was the one with relative shorter exposure, and we have pictures with longer exposure (as shown in below Figure R13) which showed similar results.

Although protection efficiency of site 1 and site 4 was higher than that of site 2, digestion of site 2 probe was obviously decreased after incubation with KSRP (in red box). Since the digestion efficiency of site 2 probe per se is not as high as that of site 1 or site 4 probe, protection of site 2 seems less significant than the other two sites.

Figure R13. RNaseH protection results with longer exposure.

Fig 5h, GFP signal needs to be quantified; could also perform flow cytometry. How do authors control for differences in transfection efficiency among constructs?

Response: As the reviewer reminded, to control the transfection efficiency among different constructs, a RFP plasmid was co-transfected simultaneously. Therefore, RFP-positive cells could represent successfully transfected cells. As a result, GFP signal was quantified based on RFP signal using Image-Pro Plus 6.0 software to show the relative intensity of GFP expression in transfected cells. As shown below (also see Figure. 3i), Y axis represents % of GFP signal in RFP-positive cells.

Figure 6i

Figure R14. Images of GFP and RFP fluorescence in 293T cells used in the in vivo processing assay.

Fig 6, lenti-GFP is not a good control for lenti-miR-129 or lenti-Zip-129.

Response: We apologized for the confusion in the control vector description. The vectors used for constructing miR-129 overexpression and knockdown lentivirus are pMIRNA1 and pmiRZip lentivector, respectively (see below Figure R15). Both vectors could express copGFP under the control of an independent promotor (EF1 promotor for pMIRNA1 and CMV promotor for pSIH-H1-copGFP shRNA Vector). Therefore we used the word "lenti-GFP" to refer to both empty vectors for convenience, which is not very accurate. Actually little difference was found between the two control groups. Additionally, in the former Fig 6c, d, e only one control group was shown because of limited space. To clarify this, we revised "lenti-GFP" to "lenti-control" or "miRZip-control" respectively. Please see the revised Figure 6, 7, S6 and S7.

Figure R15. The map of pMIRNA1 and pmiRZip lentivectors.

minor comments:

Fig1a - differentiated cells don't look like Granulocytes which typically have lobed/segmented nuclei

Response: We found that most typical segmented or band cells could be observed at day 20 of induction, however, only about 15% of total cells were segmented or band cells at day 15 (Fig. 2f, 6c and 6m). Therefore, those typical cells with lobed/segmented nuclei were scattered in one slide. We have provided another picture from the same slide at day 10 and day 15. It could be found that at day 15, 1 segmented cell, 3 metamyelocytes, 1 promyelocyte and 1 myeloblast could be observed out of a total of 6 cells. Segmented and band cells could be found in other views (as shown below Figure R16) but the proportion of such cells are relatively low.

Figure R16. The May-Grunwald Giemsa staining pictures of day 15 HPCs.

The font size in some figures are frequently hard to read (too small)

Response: We have enlarged the font size in Fig 2 and Fig 6.

a few typos

Response: We have revised the typos. Please refer to the revised manuscript with red font tags.

Reviewers' Comments:

Reviewer #1:

Remarks to the Author:

The authors have addressed my concerns. The new results have been incorporated to the revised manuscript and I am happy to support publication of their study.

Reviewer #2:

Remarks to the Author:

In this revised manuscript, the authors provided new experimental results in the attempt to address the comments made previously. However, there are still a few remaining points needed to be further clarified.

1. In the newly provided Fig.S2a, the authors further demonstrated that the level of KSRP in CD34+ HPCs was already significantly downregulated at day 3 during monocytic differentiation while remained unaltered during granulocytic differentiation. However, at the same time point (3d) in Fig.2a, abundant KSRP was shown in cells during monocytic differentiation but could be barely detected in cells under granulocytic differentiation. I understand the exposure time and experiment conditions were not the same and could make the difference in terms of the signals detected by WB analysis. However, considering the signals of GAPDH controls displayed in these studies all looked very comparable (again presumably under very different exposure time), the differences in KSRP levels in the same cells at the same time point were just too obvious to be ignored and at the same time could be quite confusing and misleading. To this end, I felt a detailed description about the numbers of cells and the exposure time used for these studies should be included in the supplementary material and method section for further clarification.

2. In the new Fig.6b, the authors have replaced the left panels related to GFP expression in cells under monocytic differentiation with better results obtained from adjusted new experimental conditions. However, the right panel of GFP expression in cells under granulocytic differentiation remained the same one from the initial submission (which also did not look good). That panel should also be replaced as well with the new results they showed in Fig.S6b.

3. By co-transfecting miR-129 inhibitor plus si-Runx1 in THP-1 and NB4 cells followed by monocytic or granulocytic differentiation, the authors performed the rescue experiment as suggested in my previous comment. However, the differences in Runx1 levels between different groups were quite small except the ones treated with si-RUNX1 alone (Fig.S8a). In particular, in NB4 cells, the 129 inhibitor group and the rescue group seemed to exhibit very similar RUNX1 expression levels. Therefore, despite that the miR-129 inhibitor-stimulated CD14 increase as well as CD11b/CSF3R reduction did seem to be sufficiently rescued by si-Runx1 co-transfection, it is hard to believe that the marginal differences (even if the difference is statistically significant) in Runx1 amounts would have such a strong biological impact as shown in Fig.S8c. How many times the WB analysis of Runx1 in different groups were performed? Was this the largest difference that could be detected?

Reviewer #3:

Remarks to the Author:

About reviewer #1

Fig R1. Authors show KSRP mRNA; reviewer asked for protein levels. Technically, sorting of monocytes did not work well. Considering the biological variability, three replicates appear not to be enough. This makes me wonder if the other experiments using human primary CD34+ cells

within this manuscript were repeated with biological replicates?

About reviewer #2

Fig R4. In miRZip-129 transduced cultures, why would the GFP-ve population express more miR-129 than in miRZip-controls?

About reviewer #3

Fig R7. The resolution of microscopy used here is poor. The authors refer to and counted "proximity signals (dots) per nucleus." Indeed, typically, PLA results in punctate fluorescent foci, but "dots" are not visible on image provided. Thus, it is not clear how authors used BlobFinder software to quantify dots per cell? How many cells were counted for each sample? Authors should have performed 3-D deconvolution since that will also show that interaction is clearly occurring within nucleus. Importantly, it would have been better if authors could detect endogenous KSRP, Drosha and Dgcr8 instead of using co-transfection to over-express them. Therefore, at the moment, authors cannot conclude that interactions occur in vivo.

Fig R12. The figure is poor quality. One should not simply take a screen shot of genome browser. The RNA-seq coverage track as shown does not support the existence of the 705 nt band labeled as pri-miR-129 on Northern blot (Fig. R11).

Fig 3H – how can primers differentiate between primary and precursor miRNA?

Response to the referees

Reviewers' comments:

Reviewer #1 (Remarks to the Author):

The authors have addressed my concerns. The new results have been incorporated to the revised manuscript and I am happy to support publication of their study.

Reviewer #2 (Remarks to the Author):

In this revised manuscript, the authors provided new experimental results in the attempt to address the comments made previously. However, there are still a few remaining points needed to be further clarified.

1. In the newly provided Fig.S2a, the authors further demonstrated that the level of KSRP in CD34+ HPCs was already significantly downregulated at day 3 during monocytic differentiation while remained unaltered during granulocytic differentiation. However, at the same time point (3d) in Fig.2a, abundant KSRP was shown in cells during monocytic differentiation but could be barely detected in cells under granulocytic differentiation. I understand the exposure time and experiment conditions were not the same and could make the difference in terms of the signals detected by WB analysis. However, considering the signals of GAPDH controls displayed in these studies all looked very comparable (again presumably under very different exposure time), the differences in KSRP levels in the same cells at the same time point were just too obvious to be ignored and at the same time could be quite confusing and misleading. To this end, I felt a detailed description about the numbers of cells and the exposure time used for these studies should be included in the supplementary material and method section for further clarification.

Response: We appreciated the reviewer for his/her suggestions. The big difference between KSRP levels at day 3 of granulocytic differentiation is indeed due to the exposure time. As shown in below Figure R1, when the exposure time reduced from 3s to 300ms, KSRP signal was significantly reduced at 0 h, 36 h and 72 h of granulocytic induction. Here, we didn't show different exposure time pictures for monocytic differentiation, because the shorter exposure (less than 3 s) would have no signals at 36 h and 72 h in that case. As the reviewer suggested, we have provided the exposure time information regarding to each Western blot result in Supplementary material and method.

In addition, we used protein quantification instead of cell number quantification for Western blot analysis in this study. Generally, a total of 20 ug protein was loaded (may vary for ± 5 ug for the sake of consistent GAPDH) for each experiment no matter of cell numbers. In this case, signal of GAPDH is very strong and milliseconds exposure time is enough therefore little difference could be detected among different exposure time.

Figure R1. Immunoblot of KSRP expression at 0 h, 36 h and 72 h of CD34⁺ HPC granulocytic differentiation at different exposure time.

2. In the new Fig.6b, the authors have replaced the left panels related to GFP expression in cells under monocytic differentiation with better results obtained from adjusted new experimental conditions. However, the right panel of GFP expression in cells under granulocytic differentiation remained the same one from the initial submission (which also did not look good). That panel should also be replaced as well with the new results they showed in Fig.S6b.

Response: We appreciated for the reviewer's nice reminding. As the reviewer suggested, we have revised Fig. 6a, 6b and 6n by replacing with the new results.

Figure 6a

Figure 6n

Figure 6b

Figure R2. The revised Fig. 6a, 6b and 6n

3. By co-transfecting miR-129 inhibitor plus si-Runx1 in THP-1 and NB4 cells followed by monocytic or granulocytic differentiation, the authors performed the rescue experiment as

suggested in my previous comment. However, the differences in Runx1 levels between different groups were quite small except the ones treated with si-RUNX1 alone (Fig.S8a). In particular, in NB4 cells, the 129 inhibitor group and the rescue group seemed to exhibit very similar RUNX1 expression levels. Therefore, despite that the miR-129 inhibitor-stimulated CD14 increase as well as CD11b/CSF3R reduction did seem to be sufficiently rescued by si-Runx1 co-transfection, it is hard to believe that the marginal differences (even if the difference is statistically significant) in Runx1 amounts would have such a strong biological impact as shown in Fig.S8c. How many times the WB analysis of Runx1 in different groups were performed? Was this the largest difference that could be detected?

Response: Previously the rescue experiment was repeated twice and similar Western results were obtained. As the reviewer pointed out, the differences in Runx1 levels between different groups were quite small. We speculated that the transfection was not efficient since THP-1 and NB4 are difficult to be transfected. To increase the transfection efficiency, Neon transfection system was used to transfect miR-129 inhibitor (or control) plus si-Runx1 (or control). As shown in the revised Fig. S8a, more obvious changes of RUNX1 protein could be observed. Accordingly, si_RUNX1 transfection significantly rescued the increased number of CD14⁺ THP-1 cells during monocytic differentiation, while rescued the decreased number of CD11b⁺ cells during granulocytic differentiation.

Figure S8

Figure R3. The rescue assay was performed by transfecting THP-1 or NB4 cells with a combination of miR-129 inhibitor (or control) plus si-Runx1 (or control), followed by monocytic or granulocytic induction for 48 h.

Reviewer #3 (Remarks to the Author):

About reviewer #1

Fig R1. Authors show KSRP mRNA; reviewer asked for protein levels. Technically, sorting of

monocytes did not work well. Considering the biological variability, three replicates appear not to be enough. This makes me wonder if the other experiments using human primary CD34⁺ cells within this manuscript were repeated with biological replicates?

Response: We apologized for the misunderstanding on this question, and here we provide protein levels of KSRP instead of mRNA levels. As to the low efficiency in monocyte sorting from normal peripheral blood, we made the efforts as below. After sorted by CD14 microbeads, the obtained CD14 positive cells were plated into T-25 flasks and cultured for 4-6 h at 37 °C until the adherent monocytes were collected by scraping (*Blood*. 2009 Jan 15;113(3):671-4.; *J Immunol Methods*. 1986 Dec 24;95(2):273-6.). This step obviously increased the percentage of CD45⁺CD14⁺ positive cells (to average of 85%, Figure R4). Next we collected another 11 buffy coat samples. KSRP protein expression was detected by Western blot and miR-129 expression was analyzed by qPCR in the newly collected 11 samples. As expected, miR-129 expression was significantly higher in neutrophils compared with that in monocytes in all the 11 samples (Figure R4). More importantly, KSRP protein expression was also significantly higher in neutrophils compared with monocytes in 9 of the 11 subjects (except for #9 and #11), which may be a result of individual variability. In summary, the relative expression of KSRP and miR-129 was significantly higher in neutrophils.

As for the biological replicates of experiments using human primary CD34⁺ cells. For each of the experiments shown in this study involving CD34⁺ primary cells, a mixture of CD34⁺ cells from 3-5 persons was used because the number of CD34⁺ cells were low in umbilical blood (≈ 100 ml each). From this collection we could obtain a total of $1-2 \times 10^6$ CD34⁺ cells. Further, for each experiment we repeated 2-3 times but only one representative of these results was shown in the manuscript. For granulocytic differentiation experiments, 3 replicates were done. For monocytic differentiation experiment, two replicates were done for CD14 staining, and another 2 replicates for CD11b-CD14 double staining. Nevertheless, the results of these experiments were consistent, further demonstrating the divergent role of KSRP/miR-129/RUNX1 axis in monocytic/granulocytic differentiation. Here we show the results of all the other replicates in Figure R6 and Figure R7. Although the differentiation degree of monocytes and granulocytes may vary, the function of KSRP, miR-129 and RUNX1 was consistent in all these replicates.

Figure R4. Flow cytometry analysis of CD45⁺CD14⁺ cells before and after monocytic sorting.

Figure R5. Analysis of KSRP protein and miR-129 levels in monocytes and granulocytes from normal human peripheral blood. In each subject, the expression of miR-129 in granulocytes was normalized to that in monocytes.

1st replicate

2nd replicate

Figure R6. A total of 2 independent replicates for functional analysis of KSRP, miR-129 and RUNX1 during granulocytic differentiation of CD34+ hematopoietic progenitor cells.

1st replicate for CD14 staining

2st replicate for CD14 staining

1st replicate for CD11b-CD14 staining

Figure R7. A total of 3 independent replicates for functional analysis of KSRP, miR-129 and RUNX1 during monocytic differentiation of CD34+ hematopoietic progenitor cells.

About reviewer #2

Fig R4. In miRZip-129 transduced cultures, why would the GFP-ve population express more miR-129 than in miRZip-controls?

Response: We appreciated the reviewer for the kind reminding. Although it seems that GFP negative population expressed more miR-129 in miRZIP-129 transduced group than that in miRZip transduced controls, no statistical difference was found (Figure R8). We have also added the *p* value to the supplementary Figure S6a and S6b.

Figure R8. CD34⁺ HPCs were transduced with lenti-129 or lenti-control, or miRZIP-129 or miRZIP-control for 24 h, and then cultured for 15 days to allow cells to differentiate into monocytes (S6a) or granulocytes (S6b). Relative miR-129 expression in GFP⁺ and GFP⁻ population from miRZIP-129 or miRZIP-control transduced HPCs were analyzed by qPCR on day 15.

About reviewer #3

Fig R7. The resolution of microscopy used here is poor. The authors refer to and counted "proximity signals (dots) per nucleus." Indeed, typically, PLA results in punctate fluorescent foci, but "dots" are not visible on image provided. Thus, it is not clear how authors used BlobFinder software to quantify dots per cell? How many cells were counted for each sample? Authors should have performed 3-D deconvolution since that will also show that interaction is clearly occurring within nucleus. Importantly, it would have been better if authors could detect endogenous KSRP, Drosha and Dgcr8 instead of using co-transfection to over-express them. Therefore, at the moment, authors cannot conclude that interactions occur in vivo.

Response: We thank the reviewer for the valuable suggestion. We understand that most results of PLA showed punctate fluorescent foci. However, dots are hard to be observed when the proximity signals are in nuclei and are extremely intense (*Proceedings of the National Academy of Sciences of the United States of America* 107, 13318-13323 (2010); *Cancer Cell*. 2015 12;27(1):72-84.). I think the major problems of our results is 1) convolution; 2) low magnification of object lens (40 X) without oil; and 3) loss of pixel quality due to picture compression.

To make the former pictures more clear, we performed 3-D deconvolution using 3D doctor software. Much more clear signals were shown in the shape of dots and the dots were

mainly found within the nucleus (Figure R9). The results of BlobFinder quantification were shown as below (Table R1), at least 100 cells were counted per experiment.

Figure R9. Former proximity ligation assays after deconvolution by 3-D doctor software.

To further demonstrate that KSRP and Drosha or DGCR8 are interacted *in vivo*, mouse anti-human KSRP antibody (MBS246006, Mybiosource) was used, pairing with Rabbit anti-human Drosha (ab12286, Abcam) or Rabbit anti-human DGCR8 (ab191875, Abcam) antibodies. A combination of anti-KSRP antibody and normal anti-rabbit IgG was used as negative control. Pictures were taken by a Zeiss LSM780 at 63X with oil immersion, and deconvolved with Huygens Essential version 16.05 (Scientific Volume Imaging, The Netherlands, <http://svi.nl>). The typical fields and counting results in 293T and HeLa cells were shown in the new Figure 5d and S5c (summarized in Figure R10), indicating strong PLA signals in KSRP plus Drosha or DGCR8 antibodies incubated cells but not in Drosha or DGCR8 and IgG antibodies incubated cells.

Figure 5d

Figure S5c

Figure R10. Proximity ligation assays using anti-KSRP, anti-Drosha or anti-DGCR8 antibodies and deconvolved by Huygens Essential version 16.05 software.

Table R1. Counting results of BolbFinder for 293T cells.

293T	KSRP+DGCR8			KSRP+Drosha		
	1	2	3	1	2	3
Number of blobs	463	325	569	1388	273	1559
Number of nuclei	54	31	29	55	12	52
Total area of nuclei	104926	81298	72972	107815	29734	116312
Intensity of signals	16719393	9634178	22007322	51375889	9600431	60503872
	1-neg	2-neg	3-neg	1-neg	2-neg	3-neg
Number of blobs	76	17	32	No signal could be detected		
Number of nuclei	167	365	124			

Total area of nuclei	203889	597873	218737	
Intensity of signals	1641123	605090	573165	

Table R2. Counting results of BolbFinder for HeLa cells.

Hela	KSRP+DGCR8				KSRP+Drosha				
	1	2	3	4	1	2	3	4	5
Number of blobs	4088	4065	2919	1019	2274	1514	1621	5383	3456
Number of nuclei	287	389	450	202	239	156	89	313	377
Total area of nuclei	642458	659274	1180153	556200	699144	320570	293683	361662	482622
Intensity of signals	9639870	96877008	75201584	28027137	34488690	29991771	30028904	1.37E+08	79516936
	1-neg	2-neg				1-neg	2-neg		
Number of blobs	12	29				166	45		
Number of nuclei	271	132				257	110		
Total area of nuclei	309692	234907				353168	375647		
Intensity of signals	149927	389103				1496627	196525		

Fig R12. The figure is poor quality. One should not simply take a screen shot of genome browser. The RNA-seq coverage track as shown does not support the existence of the 705 nt band labeled as pri-miR-129 on Northern blot (Fig. R11).

Response: We appreciated the reviewer for his/her careful review. We first want to apologize for the unclear picture and mislabel of 'KSRP' & 'Control' group in the previous version. Now, we have exported the high-resolution picture to show the RNA-seq coverage track within ~1kb range containing the pri-miR-129-1 locus (Figure R11). As shown below, we can see from Figure R11 that most regions were covered with read peaks in both KSRP & Control groups. However, because of the technological bias (eg. PCR amplification bias in library construction) and the relative precision (high-throughput method) characteristics of RNA-seq data, the whole pri-miR-129 locus may not fill with reads as we expected. Nevertheless, the further experimental methods including Northern blot, RACE and RT-PCR, confirmed the existence of the 705 nt pri-miR-129-1 transcript.

Figure S3b

Figure R11. Visualization of RNA-seq coverage across the miR-129-1 locus, showing the 705 bp length of pri-miR-129-1.

Fig 3H – how can primers differentiate between primary and precursor miRNA?

Response: In this study, we used different methods to detect primary and precursor miRNA, respectively. For primary miRNA detection, we used oligo (dT) as RT primer just as the canonical mRNA reverse transcription, because we have known that the majority of miRNA genes are transcribed by RNA polymerase II, thus the primary miRNA transcripts contain a poly(A) tails structure (*EMBO J.* 2004 Oct 13;23(20):4051-60.). As for precursor miRNA detection, we adopted a similar "stem-loop RT-PCR" method as the canonical mature miRNA detection (*Nucleic Acids Res.* 2005;33(20):e179.). As shown below, the base stacking could improve the thermal stability and extend the effective footprint of RT primer/RNA duplex that may be required for effective RT, and the spatial constraint of the stem-loop structure may prevent it from binding double-strand genomic DNA molecules and primary miRNA transcripts. We have provided these primer sequences in the Supplementary Table S6.

Figure R12. Schematic description of precursor-miR-129 RT-PCR detection (modified from Figure 1 in *Nucleic Acids Res.* 2005;33(20):e179.).

Reviewers' Comments:

Reviewer #2:

Remarks to the Author:

In this 2nd revised manuscript, the authors have addressed almost all the issues raised previously. Just one comment left to be clarified.

Previously, I have a question as to whether the biological impact from the si-Runx1 rescue experiment could really be attributed to the marginal differences in Runx1 amounts upon si-RUNX1 treatment particularly in NB4 cells (previous Fig.S8). The authors responded by providing new data using a different transfection system (Neon) to enhance siRNA transfection efficiency. However, this really did not answer my question since the rescuing effects on the % of CD11b+ cells in NB4 cells between the old and the new transfection methods were almost identical whereas the RUNX1 knockdown efficiencies were quite different. This new result further suggested two possibilities. First, granulocyte differentiation in NB4 cells could be very sensitive to the expression level of RUNX1. Even very small alterations could have big impact. Secondly, si-RUNX1 might have off-target effects. To this end, even when the level of RUNX1 did not change too much upon receiving siRNA, other genes that are potentially regulated by si-RUNX1 could be responsible for the rescue phenotype.

Reviewer #3:

Remarks to the Author:

The authors made a great effort to improve their manuscript.

REVIEWERS' COMMENTS:

Reviewer #2 (Remarks to the Author):

In this 2nd revised manuscript, the authors have addressed almost all the issues raised previously. Just one comment left to be clarified.

Previously, I have a question as to whether the biological impact from the si-Runx1 rescue experiment could really be attributed to the marginal differences in Runx1 amounts upon si-RUNX1 treatment particularly in NB4 cells (previous Fig.S8). The authors responded by providing new data using a different transfection system (Neon) to enhance siRNA transfection efficiency. However, this really did not answer my question since the rescuing effects on the % of CD11b+ cells in NB4 cells between the old and the new transfection methods were almost identical whereas the RUNX1 knockdown efficiencies were quite different. This new result further suggested two possibilities. First, granulocyte differentiation in NB4 cells could be very sensitive to the expression level of RUNX1. Even very small alterations could have big impact. Secondly, si-RUNX1 might have off-target effects. To this end, even when the level of RUNX1 did not change too much upon receiving siRNA, other genes that are potentially regulated by si-RUNX1 could be responsible for the rescue phenotype.

Response: Thank you very much for your suggestions. We also noticed that CD11b positive cells in NB4 cells were almost unchanged after the Neon transfection system was used. We agree with your analysis that off-target effect of si-RUNX1 may exist, but it is more likely that granulocyte differentiation in NB4 cells is too sensitive to cover the modulation of RUNX1 protein level. We have discussed this issue in the results section.